# Impeding pathways of intrinsic resistance in *Escherichia coli* confers antibiotic sensitization and resistance proofing

Manasvi Balachandran, Rohini Chatterjee, Ishaan Chaudhary, Chinmaya Jena, Nishad Matange[ID]*

Department of Biology, Indian Institute of Science Education and Research, Pune, Maharashtra, India

* nishad@iiserpune.ac.in

## Abstract

Pathways of intrinsic resistance in bacteria are promising targets for novel antibiotics and resistance breakers. Here, we used a genome-wide screen to identify single gene knockouts of *Escherichia coli* that were hypersusceptible to trimethoprim and chloramphenicol, two chemically diverse broad-spectrum antibiotics. Among the hits from our screen, knockouts of *acrB*, an efflux pump, and *rfaG* or *lpxM*, both involved in cell envelope biogenesis, were hypersensitive to multiple antimicrobials and could sensitize genetically resistant *E. coli* strains to antibiotics. Using experimental evolution under trimethoprim pressure, we show that high drug selection regimes drove these knockouts to extinction more frequently than wild type. Among them, Δ*acrB* was most compromised in its ability to evolve resistance, establishing it as a promising target for "resistance proofing." At a sub-inhibitory trimethoprim concentration, however, all three knockouts adapted to the antibiotic and consequently recovered from hypersensitivity, albeit to different extents. This recovery was driven by mutations in drug-specific resistance pathways, rather than compensatory evolution, frequently involving upregulation of the drug target. Notably, resistance-conferring mutations could by-pass defects in cell wall biosynthesis more effectively than efflux even though resistant mutations did not directly engage either pathway. Since inhibiting drug-efflux emerged as a better strategy, we tested the ability of chlorpromazine, an efflux pump inhibitor (EPI), to resistance proof *E. coli* against trimethoprim. While qualitatively similar in the short term, genetic and pharmacological inhibition differed dramatically on an evolutionary time scale due to evolution of resistance to the EPI. Further, adaptation to the EPI-antibiotic pair also led to multidrug adaptation. The lack of concordance between genetic and pharmacological inhibition revealed a crucial lacuna in our understanding of the mutational repertoires that facilitate adaptation to antibiotics in bacteria. We propose that while intrinsic resistance mechanisms are effective targets for antibiotic sensitization, rapid evolutionary recovery may significantly limit their utility.

**Data availability statement:** The authors confirm that all data underlying the findings are fully available without restriction. Raw data from study is provided in the S1, S2, and S3 Files. Genome sequencing data have been uploaded to Genbank (PRJNA1266351).

**Funding:** This research was funded by the DBT/Wellcome Trust India Alliance (IA/I/20/2/505181 to NM). CJ is a recipient of Senior Research Fellowship from the Council for Scientific and Industrial research (CSIR), Government of India. The funders had no role in study design, data collection and analysis, decision to publish, or preparation of the manuscript.

**Competing interests:** The authors have declared that no competing interests exist.

**Abbreviations:** DHFR, dihydrofolate reductase; EPI, efflux pump inhibitor; FIC, fractional inhibitory concentration; LB, Luria-Bertani.

## Introduction

Gram-negative bacterial infections represent a substantial public health challenge and are some of the most frequently contracted hospital-borne infections the world over. Resistance to antibiotics has further complicated their management. This is particularly true in Low- and Middle-Income Countries that have high burdens of multidrug-resistant bacteria [1]. In India, for instance, 50%–80% of hospital isolates of *Escherichia coli* and *Klebsiella pneumoniae* in the year 2021 were resistant to beta-lactams, fluoroquinolones, or cephalosporins [2]. The high prevalence of antibiotic resistance in gram-negative pathogens is attributed to horizontally acquired antibiotic-resistance genes, in addition to multiple intrinsic resistance mechanisms, such as an outer membrane permeability barrier and chromosomally encoded efflux pumps [3,4]. A multipronged approach, including the development of novel antibiotics and revitalizing existing therapeutics, is needed to tackle the AMR crisis in this group of bacteria.

The use of resistance-breakers or adjuvants that can sensitize bacteria to antibiotics is a promising strategy to reuse/repurpose antibiotics [5–7]. For instance, clavulanate and tazobactam, which are beta-lactamase inhibitors, expand the spectrum of activity of beta-lactam antibiotics [8]. More recently, intrinsic mechanisms of resistance that regulate the entry, accumulation, or metabolism of antibiotics are being explored for the design of adjuvants [3,5–7,9–11]. Genetic determinants of intrinsic antibiotic resistance, termed the "intrinsic resistome," have been identified using genome-wide screens in organisms such as *E. coli* [12–14] and *Pseudomonas aeruginosa* [15,16]. Several common themes emerge from these studies, such as the role of cell envelope, efflux proteins, and porins in regulating intrinsic drug sensitivity. Targeting these pathways has the added benefit of sensitizing bacteria to multiple classes of antibiotics simultaneously [17]. Indeed, chemical inhibitors of intrinsic resistance mechanisms, for instance, bacterial efflux pump inhibitors (EPIs), such as piperine, chlorpromazine, and verapamil enhance the antibacterial activity of antibiotics in several bacterial species [18–22]. Similarly, membrane permeabilizers including small molecules, polymyxins, and antimicrobial peptides frequently show synergy with other antibiotics, demonstrating their effectiveness as augmentative strategies against gram-negative pathogens [23–26]. Such approaches are likely to have great value, given a relative slowdown in the discovery of new antibiotics over the last decade [27].

Despite increasing promise, few studies address the evolutionary adaptive responses of bacteria to perturbations in intrinsic resistance pathways. Further, the potential of targeting intrinsic resistance mechanisms as resistance-proofing strategies, i.e., preventing de novo resistance evolution [28], remains to be explored. A few studies that investigate these aspects have focused on antibiotic-bacterium pairs where mutations directly modifying pathways of intrinsic resistance, such as altering efflux pumps expression, are a major mechanism selected under antibiotic pressure. For instance, chloramphenicol and tetracycline resistance in *E. coli* [29,30], ciprofloxacin resistance in *E. coli* [31], *Staphylococcus aureus* [32], and *P. aeruginosa*

[33], and norfloxacin resistance in *Mycobacterium smegmatis* [34], rely on up-regulation of efflux pump expression by mutations in transcriptional regulators. Eliminating or inhibiting the target efflux pumps limits or reduces the burden of resistance under antibiotic pressure in all these cases. Yet, it is unclear whether similar interventions would work against antibiotics that don't necessarily engage intrinsic resistance pathways during adaptive evolution of resistance. Further, it is unclear how the different mechanisms of intrinsic resistance compare in their ability to "resistance proof" antibiotics.

In this study, we use a genome-wide screen with trimethoprim and chloramphenicol to identify gene knockouts in *E. coli* that confer hypersensitivity to one or both antibiotics. Trimethoprim is an anti-folate that is used for broad-spectrum treatment and prophylaxis, while chloramphenicol is a protein synthesis inhibitor that is used mainly as a reserve antibiotic for severe bacterial infections. Since both are chemically distinct and target different intracellular pathways, we reasoned that a comparison of these two screens would yield drug-specific and drug-agnostic gene targets. Using hits from these screens, we ask whether intrinsic resistance pathways, i.e., drug-agnostic gene targets, can be used to sensitize drug-resistant bacteria and evaluate their ability to resistance-proof antibiotics. We use laboratory evolution of trimethoprim as a model for the latter question since resistance-conferring mutations for this antibiotic are well worked out for *E. coli* [35,36]. Being a competitive inhibitor of dihydrofolate reductase (DHFR), the most commonly encountered resistance mutations map to the *folA* locus [36–38]. In addition, mutations in *mgrB*, a feedback regulator of PhoQP signaling, also occur frequently in trimethoprim-resistant *E. coli* [38,39]. By tracking these mutational signatures, we show that evolutionary recovery from antibiotic hypersensitivity occurs to varying extents across antibiotic hypersensitive genetic backgrounds, demonstrating that certain mechanisms for antibiotic sensitization are more effective than others.

## Results

### Cell wall biosynthesis, membrane transport, and information transfer pathways regulate intrinsic trimethoprim resistance in *E. coli*

We screened the Keio collection of *E. coli* knockouts [40], which is a library of ~3,800 single-gene deletions, for mutants that were hypersensitive to trimethoprim or chloramphenicol. Knockout strains were grown in LB media supplemented with antibiotics at their respective IC50 values or without antibiotic (control). Optical density at 600 nm was averaged across duplicate measurements for each knockout strain expressed as fold over wild type (Fig 1A). A resultant Gaussian distribution of drug susceptibilities with a mean ~1 (Fig 1B and S1 File) was obtained. Knockouts that showed poor growth in the presence of antibiotic, i.e., lower than two standard deviations from the median of the distribution, but not control media, were classified as hypersensitive (Fig 1A and 1B). This resulted in 35 and 57 knockouts that were hypersensitive to trimethoprim or chloramphenicol, respectively (S1 File). An enrichment of genes involved in cell envelope biogenesis, information transfer, and membrane transport pathways (classified using the Ecocyc database [41]) was evident in both data sets (Fig 1C and S1 File). While antibiotic-specific knockouts, such as *nudB* (folate metabolism) for trimethoprim and *rplA* (ribosomal protein) for chloramphenicol were identified, some hypersensitive mutants were common between the two screens, particularly those regulating LPS synthesis or drug efflux (Fig 1D and S1 File).

To validate the results of our screen, we analyzed the growth of all 35 trimethoprim hypersensitive strains on solid media supplemented with MIC, MIC/3, and MIC/9 of trimethoprim. Two of the 35 hypersensitive strains did not revive satisfactorily from frozen stocks and were not used for subsequent analysis. Close to two-thirds of the hits (20/33) showed compromised colony formation on trimethoprim-supplemented agar as well (Fig 2 and S2 File). Some hits from the chloramphenicol screen also showed hypersensitivity to trimethoprim, particularly those involved in LPS biosynthesis (Fig 2 and S2 File). The highest sensitization to trimethoprim was observed for knockouts of genes directly involved in folate metabolism, such as *nudB*, or for known regulators of intrinsic resistance to antibiotics, such as *acrB* and the *rfa* operon (Fig 2 and S2 File). Thus, our screen identified both drug-target specific and target-agnostic determinants of antibiotic hypersensitivity in *E. coli.*

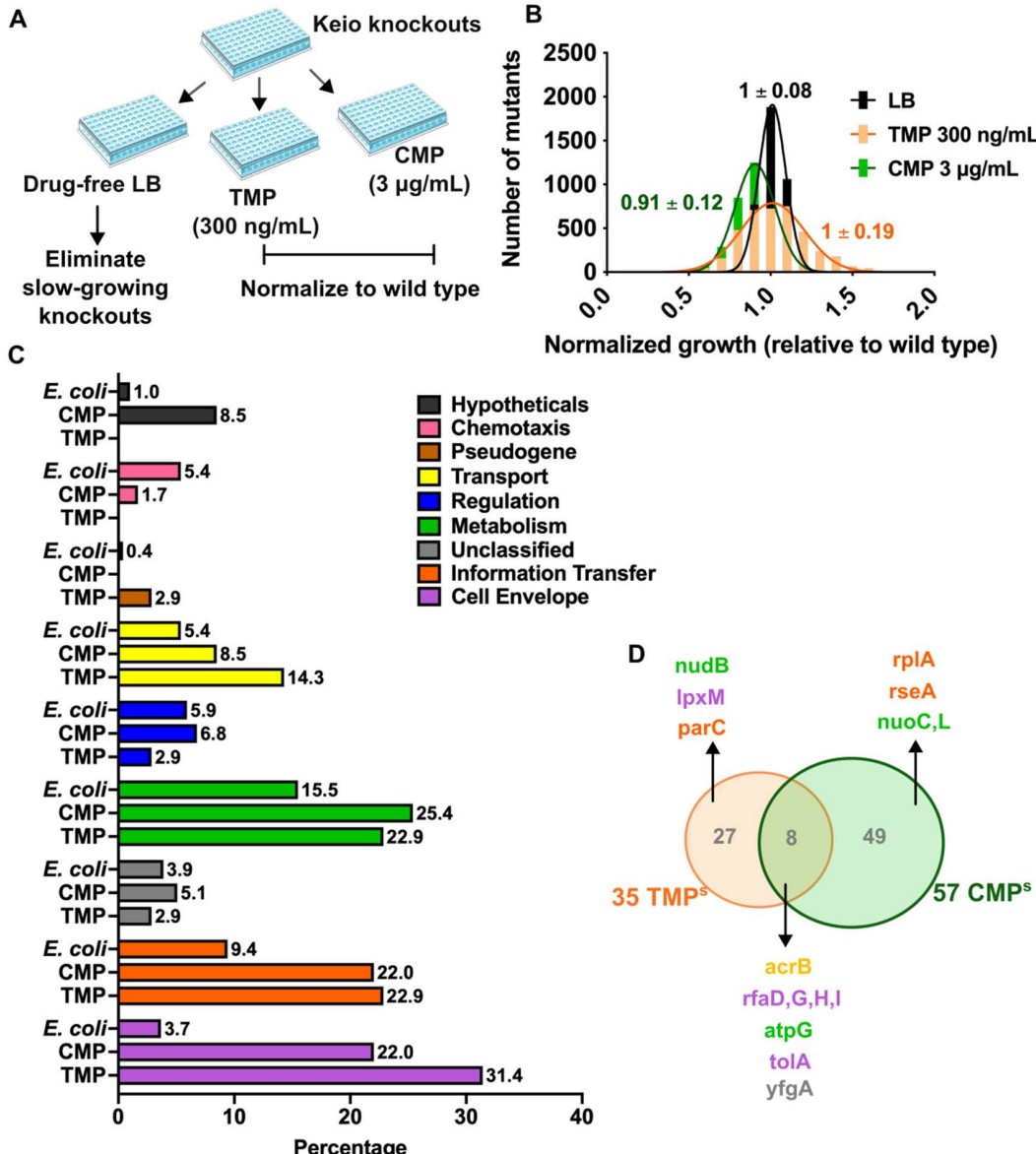

**Fig 1. Genome-wide screen to identify *Escherichia coli* gene knockouts hypersensitive to trimethoprim and chloramphenicol. (A)** Schematic showing the screening strategy for identifying hypersensitive gene knockouts from the Keio mutant library. All mutants were cultured in LB and LB supplemented with trimethoprim (TMP) and chloramphenicol (CMP) at their respective IC50s. **(B)** Distribution of drug-susceptibilities of Keio mutants grown in indicated growth media obtained after normalizing the growth of each mutant to wild type (set to 1). Mean ± S.D. derived after fitting each distribution to a Gaussian function are shown. **(C)** Analysis of gene functional categories for hypersensitive mutants compared with the frequency of genes in each category encoded by *E. coli* K-12 MG1655. Percentage of the total in each case is provided above each bar. **(D)** Comparison of hypersensitive mutants identified in the trimethoprim and chloramphenicol screens shown as a Venn diagram. Representative genes from each set are named and gene names are colored by functional categories consistent with panel B. The data underlying this Figure can be found in S1 File.

## Modulators of intrinsic resistance can sensitize antibiotic-resistant *E. coli*

We next asked whether hits from our genetic screen could induce hypersensitivity to a wider range of antibiotics. We chose 3 knockouts from established intrinsic resistance pathways for these analyses, namely, *acrB*, which codes for part

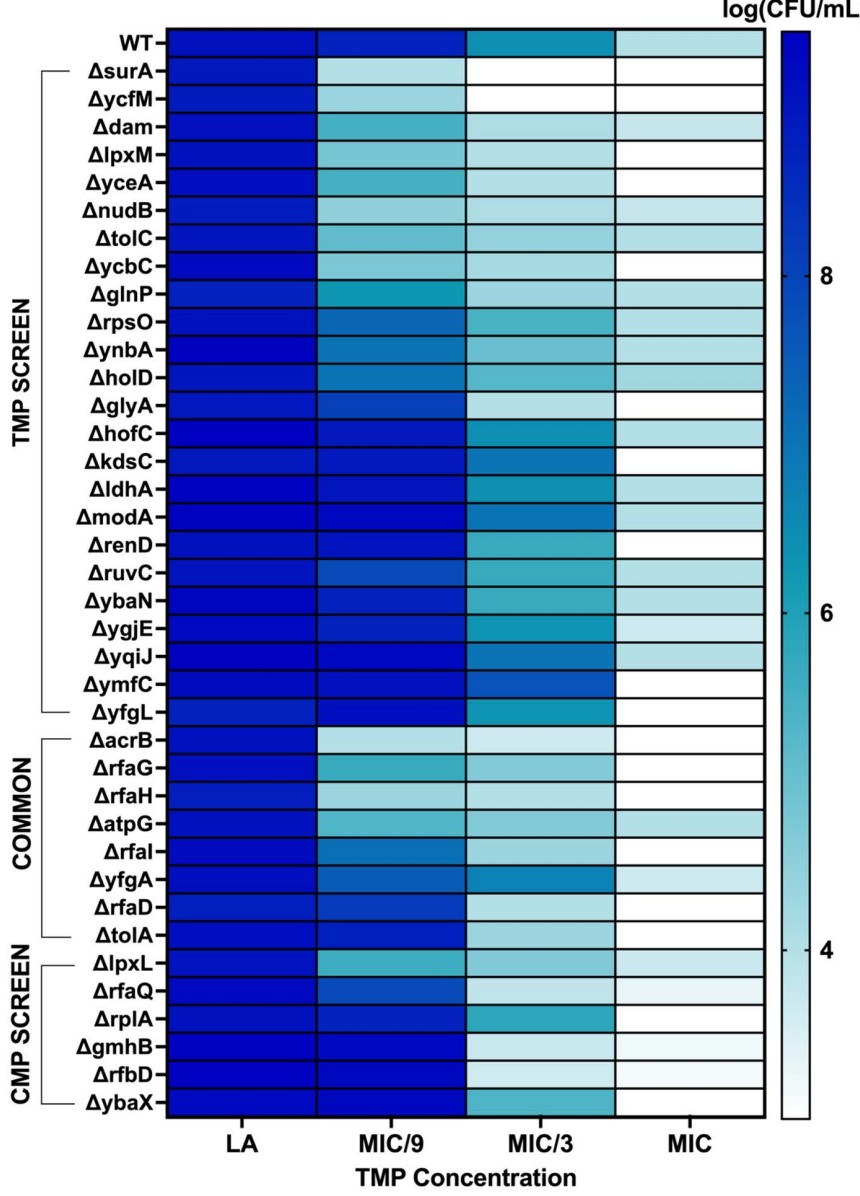

**Fig 2. Colony-forming efficiencies of *Escherichia coli* K-12 BW25113 and hypersensitive mutants from the trimethoprim and chloramphenicol screens on drug-free or trimethoprim-supplemented growth media.** Three different concentrations of trimethoprim corresponding to MIC, MIC/3, and MIC/9 for the wild type were used. Log(CFU/mL) values are shown as a color gradient. Data represent the mean from three independent replicates. The data underlying this Figure can be found in S2 File.

of the AcrAB-TolC multidrug efflux pump, *rfaG,* which codes for lipopolysaccharide glucosyl transferase I and *lpxM*, which codes for Lipid A myristoyl transferase. For all three knockouts, we expected hypersensitivity due to greater intracellular concentrations of the antibiotic as a results of reduced efflux (in the case of *acrB*) or higher permeation into the cell due to cell envelope perturbation (in the case of *lpxM* and *rfaG*). We also included *nudB* (dihydroneopterin triphosphate diphosphatase) as a representative trimethoprim-specific hypersensitive strain which would result in impaired folate biosynthesis. Each of these gene deletions was first introduced into *E. coli* K-12 MG1655 to ensure a clean genetic background. We

then tested the susceptibility of the 4 knockouts against sub-MIC concentrations of trimethoprim:sulfamethoxazole (clinically relevant formulation of trimethoprim), chloramphenicol, erythromycin, spectinomycin, amoxicillin, piperacillin, nalidixic acid, rifampicin, or colistin (Fig 3A and S2 File). These antibiotics include major classes currently used against gram-negative bacteria. All four knockouts were hypersensitive to trimethoprim:sulfamethoxazole, while all except Δ*nudB* were hypersensitive to chloramphenicol. Knockouts of *acrB*, *lpxM*, and *rfaG* were hypersensitive against the other antibiotics as well, but to varying extents (Fig 3A), while *nudB* had only a marginal impact. In broth cultures too, deletion of *acrB*, *lpxM*, and *rfaG* reduced IC50 values of multiple antibiotics (Fig 3B and S2 File). Loss of *acrB* resulted in the highest magnitude of hypersensitivity against most antibiotics tested (Fig 3B).

Next, we asked whether these gene knockouts could reverse (i.e., sensitize) the phenotypes of genetically drug-resistant strains. For these analyses, we chose representative laboratory-evolved *E. coli* isolates from earlier work [42] that were resistant to trimethoprim, erythromycin, nalidixic acid, rifampicin, or amoxicillin. These isolates had clearly identifiable chromosomal mutations that are known to cause resistance (Fig 3C and S2 File). We then knocked out *acrB*, *lpxM*, or *rfaG* from each isolate and measured antibiotic IC50 values. While in most cases we observed a decrease in drug IC50 value, there was substantial variability in the extent of antibiotic sensitization that could be achieved. Once again, *acrB* deletion was the most effective at reducing resistance across antibiotics. Indeed, in some instances, such as erythromycin and nalidixic acid, deletion of *acrB* resulted in a complete reversal of resistance, and drug IC50 values were reduced to the level of the drug-sensitive *E. coli* MG1655 (Fig 3C and S2 File).

## Evolutionary recovery from trimethoprim hypersensitivity varies across drug concentration regimes

We next asked if antibiotic sensitization by perturbing intrinsic resistance mechanisms could also result in resistance proofing, i.e., preventing de novo resistance evolution [28]. Experimental evolution of trimethoprim resistance was an ideal model to address this question since the known pathways of resistance against this antibiotic are well-worked out and mainly involve mutations in drug-target or associated genes [38,39,42]. We first measured the impact of different trimethoprim concentrations on the carrying capacities of wild-type and Δ*acrB*, Δ*rfaG*, Δ*lpxM* strains after a single growth cycle. Starting at a population density of ~$2 \times 10^7$ CFU/mL all strains achieved similar densities of ~$2 \times 10^9$ CFU/mL in antibiotic-free media (Fig 4A and S2 File). In trimethoprim-supplemented media, we observed two distinct phases of antibiotic action. At low concentrations (corresponding to MIC/9 and MIC/2 for wild type), trimethoprim was bacteriostatic and prevented an increase in bacterial density of all strains, though MIC/9 trimethoprim was sufficient to induce stasis for the knockouts but not wild type (Fig 4A and S2 File). At MIC of the wild type, substantial loss in viability was observed for all genotypes, but no further decline in viability occurred at higher concentrations. Notably, high trimethoprim (i.e., MIC and above) reduced the viability of wild type by 100-fold, whereas 1,000-fold reduction was observed for all three hypersensitive knockouts (Fig 4A).

Spontaneous mutants occur at frequencies of ~1 in $10^8$ in *E. coli*, and trimethoprim can itself increase this number by 10-fold [43]. Further, our earlier work has shown that adaptation to trimethoprim is very rapid, i.e., detectable within 5 cycles of growth in sub-MIC antibiotic and often occurs at high frequencies due to the involvement of IS-elements [38,42]. We therefore performed an antibiotic challenge experiment at MIC or MIC/9 of trimethoprim for 30 replicate populations each of wild type, Δ*acrB*, Δ*rfaG*, and Δ*lpxM* strains lasting 5 growth cycles. Each population had a starting density of ~$2 \times 10^7$ CFU/mL and was serially passaged with a 1% bottleneck between growth cycles. Under these conditions, at high drug pressure, two possible outcomes were expected, i.e., most replicate populations would be driven to extinction by dilution, though a small number may survive due to selection of pre-existing resistant clones. This was indeed the case, and for wild-type *E. coli* only 8 of the 30 replicate populations survived. This number could be lowered by increasing the antibiotic dose to 2× or 4×MIC, with 1 and 0 populations surviving, respectively (Fig 4B). Knockouts of *rfaG* and *lpxM* showed slightly reduced numbers of surviving populations (4 and 3, respectively) at the MIC of the wild type. Strikingly, all replicates of Δ*acrB* were driven to extinction demonstrating a marked reduction in adaptation to trimethoprim at high drug

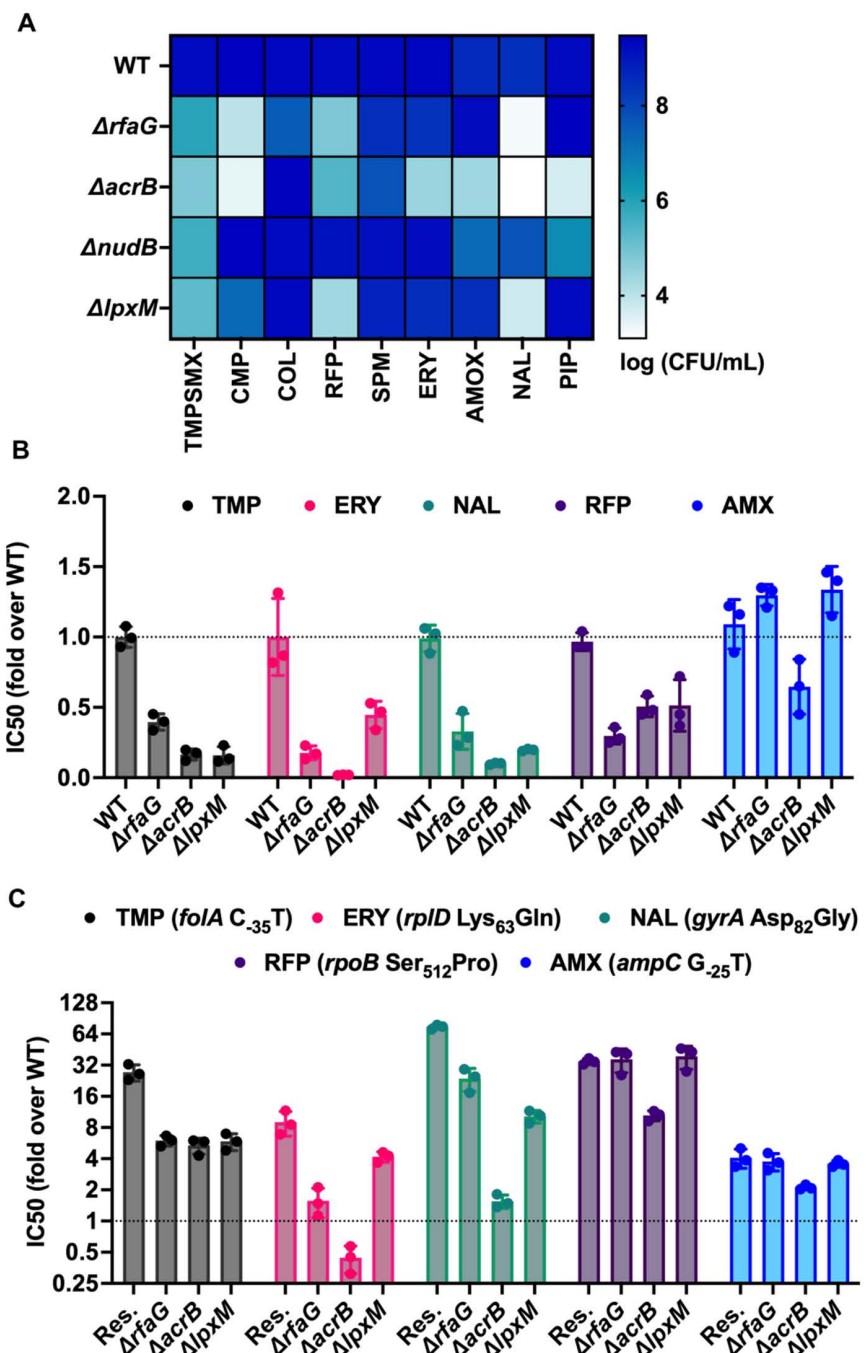

**Fig 3. Broad-spectrum antibiotic sensitization upon inhibition of intrinsic resistance pathways in *Escherichia coli*. (A)** Colony-forming efficiencies of *E. coli* MG1655 WT and knockouts in LA alone or supplemented with subinhibitory concentrations of trimethoprim-sulfamethoxazole (TMPSMX), chloramphenicol (CMP), colistin(COL), rifampicin (RFP), spectinomycin (SPM), erythromycin (ERY), amoxicillin (AMOX), nalidixic acid (NAL), and piperacillin (PIP). Log(CFU/mL) values are shown as a color gradient. A concentration of each antibiotic that showed less than one order of magnitude reduction in the growth of the wild type was selected for this assay. Log(CFU/mL) values are shown as a color gradient. Data represent mean from three independent replicates. **(B)** IC50 values of amoxicillin (AMX), rifampicin (RFP), nalidixic acid (NAL), erythromycin (ERY), and trimethoprim (TMP) for indicated gene knockouts normalized to the wild type (WT) are plotted. No change in IC50 would result in a value of 1. Mean±S.D. from three independent experiments are plotted. **(C)** Normalized IC50 values of amoxicillin (AMX), rifampicin (RFP), nalidixic acid (NAL), erythromycin (ERY), and trimethoprim (TMP) for indicated gene knockouts in drug-resistant backgrounds are plotted. A value of 1 would indicate similar IC50 as the drug-susceptible wild type. The normalized IC50 values of each resistant background (Res.) is also shown. The resistance-conferring mutation in each of the resistant strains used in this experiment are provided in the color key. Mean±S.D. from three independent experiments are plotted. The data underlying this Figure can be found in S2 File.

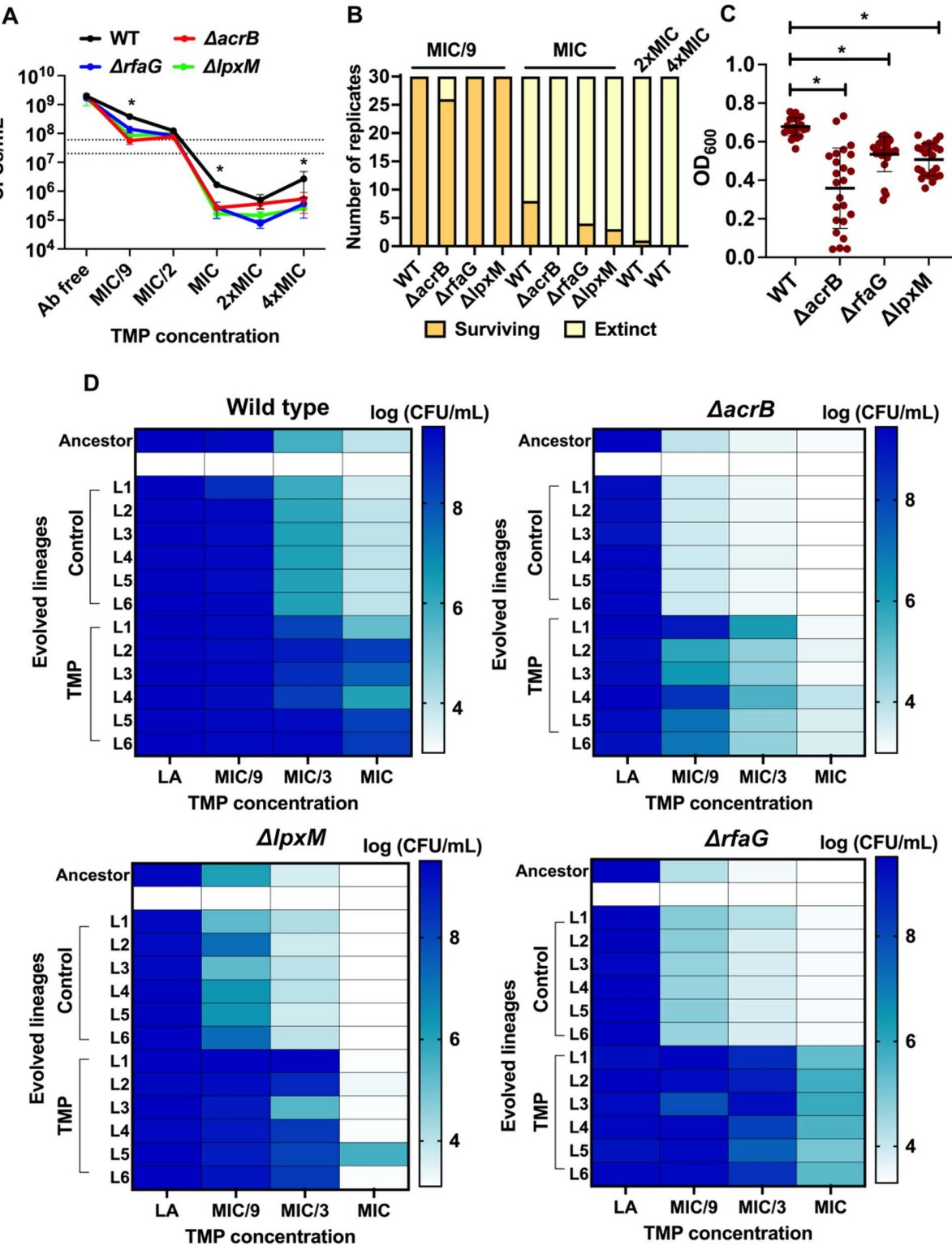

**Fig 4. Hypersensitive gene knockouts of *Escherichia coli* adapt to different extents under trimethoprim pressure. (A)** Carrying capacities of wild type (WT) and knockout strains represented as colony-forming units (CFUs)/mL at different concentrations of trimethoprim (relative to the MIC of the WT strain) after one growth cycle. Data represent mean±S.D. from three independent experiments. Statistically significant differences (Students

*t* test, *p*-value < 0.05) compared to wild type are indicated by \*. **(B)** Antibiotic challenge at MIC/9 or MIC of trimethoprim for 30 replicate populations of wild-type *E. coli* (WT) and single gene knockouts. Number of surviving populations after 5 serial transfers are plotted. Wild type was also subjected to 2× and 4×MIC trimethoprim in a similarly designed serial transfer paradigm. **(C)** Optical density at 600 nm ($OD_{600}$) of populations of indicated genotypes challenged with MIC/9 trimethoprim concentrations after 5 growth cycles are plotted. Each point represents the value of $OD_{600}$ for a single replicate and mean ± S.D. of all 30 replicates are shown. Statistically significant differences (Students *t* test, *p*-value < 0.05) compared to wild type are indicated by \*. **(D)** Colony-forming efficiencies of six lineages of the wild type and knockout strains (L1 to L6) evolved in the absence or presence of MIC/9 trimethoprim after 20 growth cycles (~140 generations of evolution). Three different concentrations of trimethoprim were used corresponding to MIC, MIC/3, and MIC/9 for the wild type to the determine extent of adaptation. Log(CFU/mL) values are shown as a color gradient. Data represent the mean from three independent replicates. The data underlying this Figure can be found in S2 File.

pressure (Fig 4B). At low trimethoprim, i.e., MIC/9, extinction by dilution was only likely if the potential for adaptation was lower, i.e., fewer genetic mechanisms of adaptation were accessible to the knockouts. All wild-type populations adapted to MIC/9 trimethoprim achieving consistently high $OD_{600}$ at the end of the fifth passage (Fig 4B and 4C). All populations of *rfaG* and *lpxM* knockouts too improved their growth over 5 days of culturing, but achieved lower $OD_{600}$ values than wild type at the end of the fifth growth cycle (Fig 4B and 4C). Once again, loss of *acrB* proved to be most detrimental for the ability of *E. coli* to adapt to trimethoprim. Unlike the other genotypes, a small number of replicates (4 out of 30) of Δ*acrB* were driven to extinction even at MIC/9 drug pressure. Moreover, most of the surviving Δ*acrB* populations showed dramatically lower $OD_{600}$ values at the end of passage 5 (Fig 4B and 4C).

Since the above results were only indicative of early time points, we performed evolution at MIC/9 for 6 replicates of each genotype for 20 growth cycles (~140 generations) and asked if longer duration of adaptation to low antibiotic could facilitate greater recovery from antibiotic hypersensitivity. Control evolution experiments in drug-free media were set up in parallel. To assess whether these adapted lineages had recovered to the same extent as wild type, we assessed their colony-forming efficiency at MIC/9, MIC/3, and MIC of trimethoprim. Wild-type *E. coli* evolved in trimethoprim, but not drug-free medium, showed increased colony formation across all three concentrations of trimethoprim (Fig 4D and S2 File). All 3 hypersensitive knockout strains also showed improved growth in the presence of trimethoprim. However, the extent of recovery from hypersensitivity varied substantially between them (Fig 4D and S2 File). All lineages of the *rfaG* knockout approached the colony-forming efficiency of drug-adapted wild type, indicating near complete reversal of hypersensitivity. On the other hand, drug-adapted Δ*acrB* continued to form fewer colonies at all concentrations of trimethoprim than the wild type, while an intermediate level of recovery was observed for the Δ*lpxM* mutant, i.e., growth comparable to wild type at MIC/3, but lower growth at MIC (Fig 4D and S2 File). Thus, though all 3 targets reduced the potential for adaptation to trimethoprim resistance they did so to varying degrees. Further, though their growth at different trimethoprim concentrations was comparable to begin with, potential for resistance evolution was significantly reduced only in the *acrB* deletion strain.

### Trimethoprim resistance-conferring mutations and phenotypic adaptations drive differential evolutionary recovery from hypersensitivity across knockout backgrounds

We reasoned that differential adaptation to trimethoprim across the hypersensitive strains was because, either, they adapted to trimethoprim by altogether different mutational trajectories, or, the effects of similar beneficial mutations differed across the three knockouts. We sequenced the genomes of control and trimethoprim-evolved populations at the final time point to distinguish between these two possibilities. Adaptive mutations were identified by comparing the genome sequences of evolved populations with respective ancestors and only those mutations occurring at a frequency of 20% or higher were considered for further analyses. For each genetic background, we then classified mutations that arose in LB alone as "*media adaptations*" and those that arose exclusively in the presence of trimethoprim as "*trimethoprim adaptations*". To ensure that piggybacking mutations did not confound our analyses, we only considered a mutation as adaptive if it occurred in at least 2 independent lineages or was in a gene with a known link to folate metabolism (S3 File).

For wild type, 4 of 6 drug-adapted lineages had mutations at the *folA* locus, which codes for DHFR, i.e., the target of trimethoprim. These were either point mutations in the gene promoter or an increase in gene copy number (Fig 5 and S3 File). Both these kinds of mutations are beneficial as they increase the expression level of DHFR [38,39,42]. In addition, 3 lineages also had inactivating mutations at the *mgrB* locus (Fig 5 and S3 File). In earlier work, we have shown that loss of MgrB activates the PhoQP two-component signaling pathway and confers resistance to trimethoprim by upregulating transcription of *folA* [38,39]. A few lineages had mutations in RpoS as well (Fig 5 and S3 File) that lead to general fitness enhancement, but do not affect trimethoprim MIC [38,39,42]. Lineages of *E. coli* Δ*acrB* evolved in trimethoprim showed a similar mutational signature as wild type, i.e., mutations at the *folA*, *mgrB*, and *rpoS* loci (Fig 5 and S3 File). Likewise, all evolved lineages derived from *E. coli* Δ*lpxM* carried mutations at *folA* and *rpoS* loci, though a few carried additional mutations in the *prlF* and *folM* genes. Mutations in *hns* were also detected, but these were not specific to evolution in tri-methoprim and thus were likely to be a media adaptation unique to this genetic background (Fig 5 and S3 File). For *E. coli* Δ*rfaG*, which showed highest recovery from hypersensitivity, only 1 of the 6 trimethoprim-evolved lineages had mutations at the *folA* locus. Mutations in *mgrB* were detected in 3 lineages, while *folD* was mutated in 1 lineage (Fig 5 and S3 File). Interestingly, 2 additional genes were repeatedly mutated in this genetic background. The first of these, *pitA*, which codes for a metal phosphate:H$^+$ symporter was a relatively infrequent mutation in earlier evolution experiments with wild-type *E. coli* [39]. The second locus, *dsbA*, which codes for a thiol:disulfide oxidoreductase chaperone, was a novel mutation that has not been reported earlier in the context of trimethoprim resistance (Fig 5 and S3 File).

We sought to understand the contribution of *pitA* and *dsbA* toward trimethoprim adaptation. Mutations in both genes included disruption of the coding sequence by IS elements (Fig 5 and S3 File). Further, one lineage had evolved a Cys$_{30}$Ser missense mutation in DsbA, which is known to inactivate the chaperone [44,45], indicating that loss of the gene's function was beneficial in trimethoprim (S3 File). In line with this prediction, deletion of *pitA* or *dsbA* enhanced growth of *E. coli* in the presence of trimethoprim (Fig 6A and S2 File). DsbA activity is required for folding of MgrB in *Salmonella*, and deficiency of either protein leads to activation of PhoQP [46]. On the other hand, PitA is itself a target of PhoQP and being a phosphate importer [47] could modulate the activation of PhoP independent of PhoQ. Therefore, we asked if acti-vation of PhoQP may explain the emergence of these mutations under trimethoprim pressure. Indeed, the activity of the *phoP* promoter, which it auto-regulated by PhoQP, was higher in *dsbA* and *mgrB* knockout strains, but not in *E. coli* Δ*pitA* (Fig 6B). Further, the expression of DHFR, which is enhanced by activation of PhoQP, was also higher in *E. coli* Δ*dsbA* and Δ*mgrB*, but not in *E. coli* Δ*pitA* (Fig 6C). Thus, loss of DsbA and MgrB conferred resistance by PhoQP-dependent upregulation of DHFR expression. Loss of PitA, on the other hand, conferred PhoQP/DHFR-independent trimethoprim resistance.

We next addressed why hypersensitive knockouts showed varying degrees of recovery upon sub-MIC trimethoprim exposure, specifically focusing on Δ*rfaG* and Δ*acrB*. Regardless of the ancestral genotype, most evolved lineages had either 1 or 2 resistance-conferring mutations (Fig 5 and S3 File). Thus, the number of adaptive mutations accumulated could not explain different extents of recovery. Since similar molecular pathways of adaptation were identified across knockout backgrounds, we wondered whether recovery from hypersensitivity was proportionate to how well a gene deletion could reduce the benefit of resistance-conferring mutations. To test this, we knocked out *rfaG* and *acrB* from a panel of 5 trimethoprim-resistant *E. coli* strains with different mechanisms of resistance. Three of these were spontaneous mutants isolated from earlier laboratory evolution experiments harboring different mutations in *folA* [38,39]. Additionally, we also tested Δ*mgrB* and Δ*pitA* to cover all major resistance pathways encountered during evolution (Fig 6D). Interest-ingly, extent of reduction in IC50 across this panel of resistant strains was significantly lower upon *rfaG* deletion compared to *acrB* (Fig 6D and 6E). These data indicated that, in general, resistance-conferring mutations could overcome the phe-notypic effects of *rfaG* deletion more effectively, thus driving reversal of hypersensitivity.

We also noticed that a few lineages of *E. coli* Δ*rfaG* showed greater colony formation in trimethoprim that could not be explained based on mutations alone (Figs 4B and 6A). This discrepancy suggested additional non-genetic adaptations in

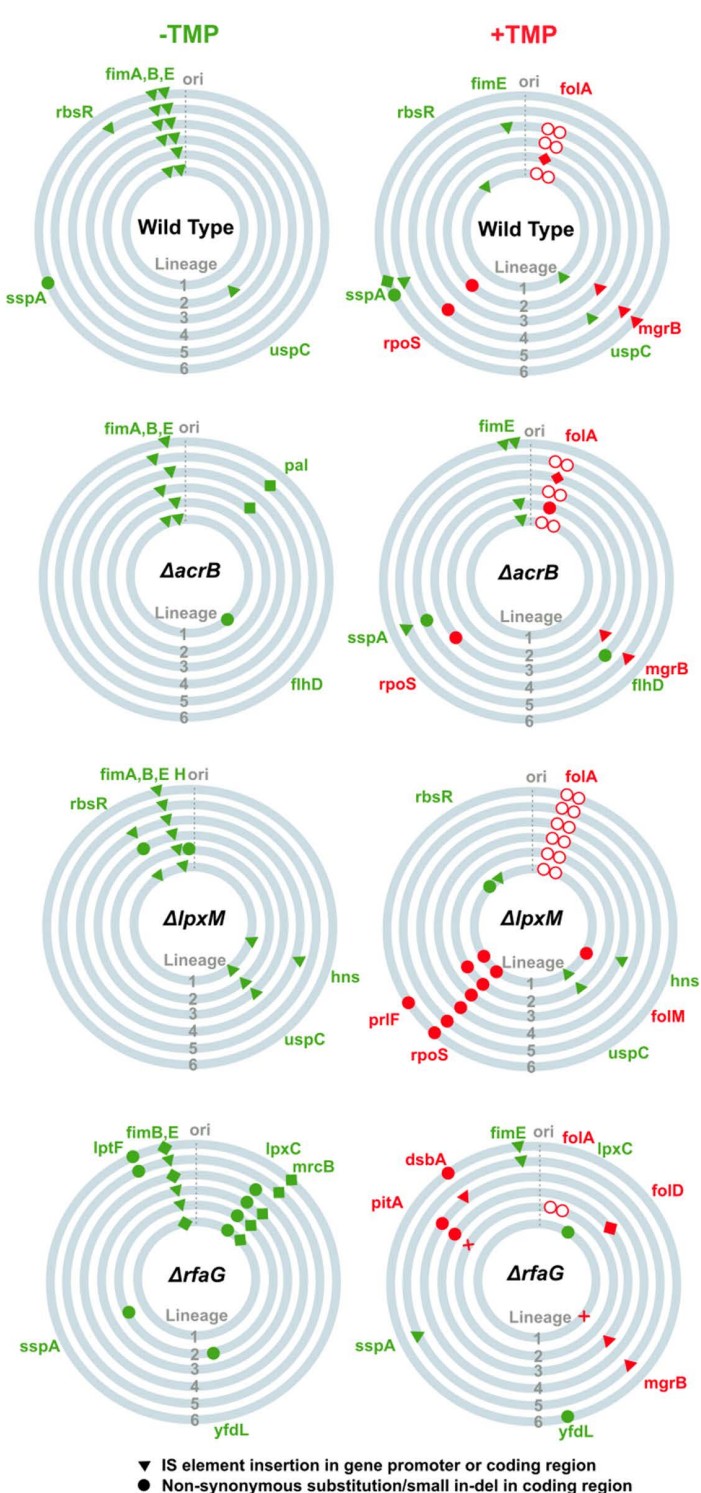

**Fig 5. Mutational landscape of six replicate lineages of *Escherichia coli* wild type and indicated gene knockouts in LB alone or low trimethoprim concentration (100 ng/mL) after ~140 generations of evolution.** Each circle represents a single lineage. Mutations in green occur during evolution in LB alone and represent media adaptations. Mutations labeled in red are those that occurred only in the presence of trimethoprim and represent drug-specific adaptations. All listed mutations occur in at a frequency of at least 20% in the population, were either detected in 2 or more lineages and/or were in genes known to be involved in folate metabolism. The data underlying this Figure can be found in S3 File.

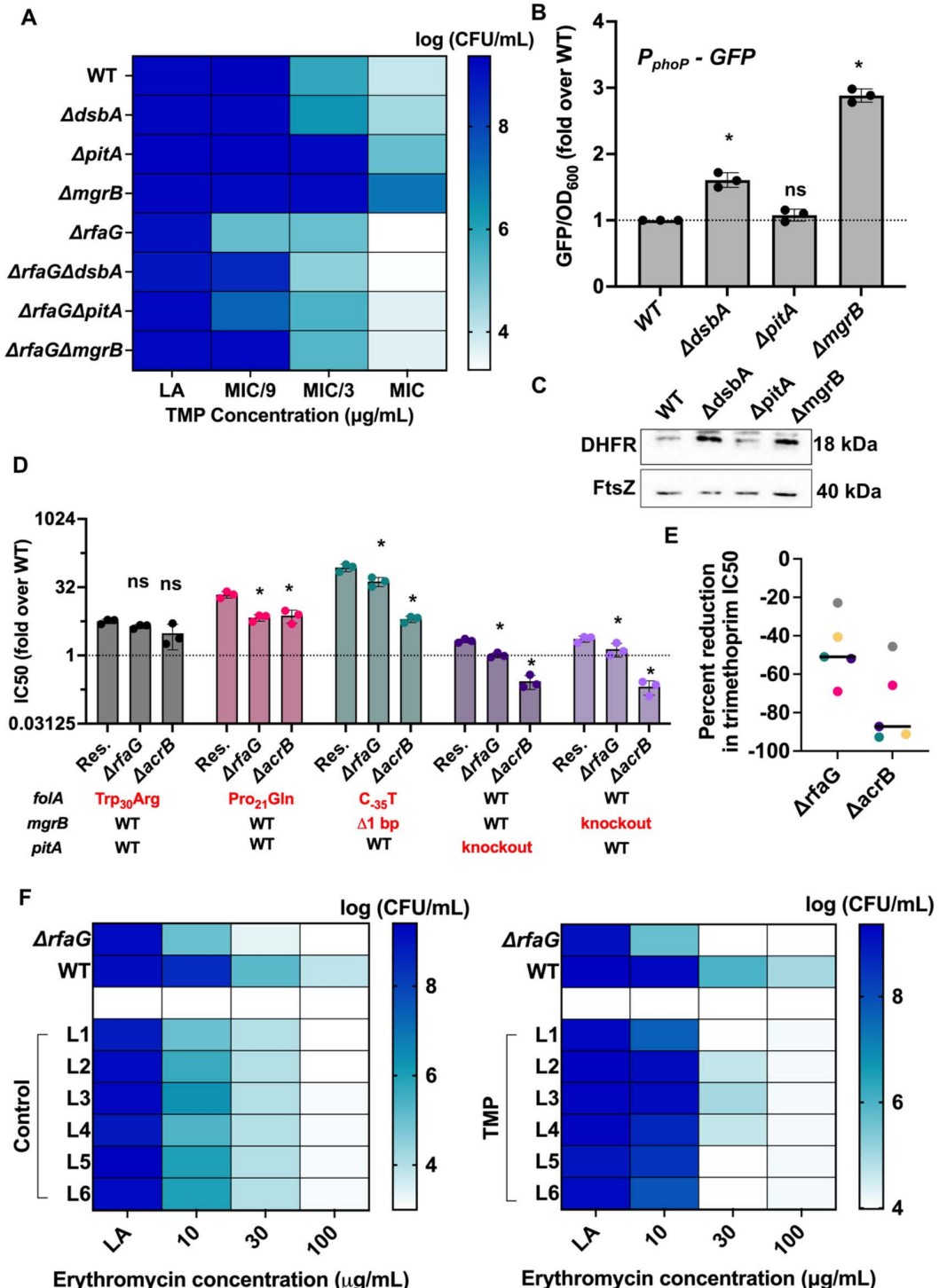

**Fig 6. Mechanisms of trimethoprim resistance facilitate different extents of recovery from a hypersensitive phenotype in *Escherichia coli* ΔrfaG and ΔacrB.** **(A)** Effect of *dsbA*, *pitA,* and *mgrB* deletion on colony-forming efficiency of *E. coli* wild type (WT) and *ΔrfaG* in the presence of tri-methoprim. Three different concentrations of trimethoprim were used corresponding to MIC, MIC/3, and MIC/9 of the wild type. The mean of log colony-forming efficiencies from three independent replicates are represented as a color gradient. **(B)** Activity of the *phoP* gene promoter in wild-type or indicated knockout strains, assessed using a GFP reporter plasmid (pUA66-$P_{phoP}$-GFP). Promoter activity was calculated as the ratio of GFP

fluorescence and optical density (at 600 nm) of the culture. Values of GFP/OD$_{600}$ for knockout strains are expressed as fold over wild type (WT). Statistical significance was tested using an unpaired *t* test. *P*-value of <0.05 (*) is indicated. **(C)** DHFR expression level was assessed in the indicated strains by immunoblotting using anti-DHFR polyclonal antibody. FtsZ was used as a loading control. Data shown are representative of three independent replicates. **(D)** Trimethoprim IC50 values of resistant *E. coli* strains (Res) and their Δ*rfaG* and Δ*acrB* knockout derivatives. Values of IC50 are expressed as fold over *E. coli* wild type. Mean ± S.D. from three independent experiments are plotted. Statistical significance was tested using an unpaired *t* test. *P*-value of <0.05 (*) is indicated. Mutations linked with trimethoprim resistance in each strain are shown below the plot. **(E)** Scatter plot of percent reduction in IC50 upon deletion of *rfaG* and *acrB* across different trimethoprim-resistant strains. Each point on the scatter represents a unique resistant genotype and is colored consistently with panel D. The two distributions were compared using a paired *t* test and the difference was statistically significant (*p*-vale < 0.05). **(F)** Phenotypic adaptation of *E. coli* Δ*rfaG* evolved in drug-free (left) or trimethoprim-supplemented media (right) to erythromycin. Data represent mean colony-forming efficiencies of evolved lineages at varying concentrations of erythromycin from three replicates. The data underlying this Figure can be found in S2 File.

the *rfaG* knockout populations. In order to test this, we assessed the colony-forming efficiency of evolved Δ*rfaG* populations on erythromycin, a hydrophobic antibiotic that the ancestral strain was hypersusceptible to. Interestingly, trimethoprim-adapted Δ*rfaG* showed greater colony formation on erythromycin as well (Fig 6F and S2 File). Thus, the ability of *E. coli* Δ*rfaG* to evolve greater resistance to trimethoprim could be attributed to a combination of phenotypic recovery under drug pressure and genotypic resistance.

## Pharmacologically induced trimethoprim hypersensitivity is an unstable phenotype over evolutionary timescales

Based on the above experiments, AcrB emerged as the most favorable target for both, trimethoprim sensitization and resistance proofing. We therefore asked whether it was possible to recapitulate these findings using a chemical inhibitor of the AcrAB-TolC pump. Chlorpromazine is an antipsychotic drug that is also a bacterial EPI and is predicted to bind to AcrB directly based on docking studies [20]. Chlorpromazine and trimethoprim synergistically/additively inhibited the growth of wild-type *E. coli* in a checkerboard assay (Fig 7A). We chose 50 μg/mL chlorpromazine for subsequent evolution experiments, since this was the lowest concentration at which additivity was seen across the entire range of trimethoprim concentrations. In combination with chlorpromazine, high trimethoprim pressure (i.e., at MIC) drove all 30 replicate wild-type *E. coli* populations to extinction within 5 cycles of growth reminiscent of the *acrB* knockout (Figs 7B and 4B). Interestingly, chlorpromazine in combination with MIC/9 trimethoprim also resulted in a majority of replicates being driven to extinction (Figs 7B and 4B), indicating that it may inhibit other efflux systems in *E. coli* as well resulting in more robust resistance proofing than the *acrB* knockout. Thus, for short durations of trimethoprim exposure, the EPI showed promising results as a resistance-proofing adjuvant. As with the *acrB* knockout (Fig 4), we continued the evolution experiment for 6 surviving lineages over 20 cycles of growth (~140 generations) to understand if over long term too, pharmacological and genetic inhibition of the AcrB similarly impeded trimethoprim adaptation. Surprisingly, very high levels of trimethoprim resistance were observed for all 6 lineages of wild type evolved in MIC/9 trimethoprim and chlorpromazine, in stark contrast to the *acrB* knockout (Fig 7C and S2 File). This increase in resistance was not observed for lineages evolved in chlorpromazine alone indicating that the adjuvant-antibiotic combination selected for higher level resistance than either drug alone (Fig 7C and S2 File).

We sequenced the genomes of chlorpromazine and chlorpromazine-trimethoprim evolved populations to decipher the underlying mechanism. Chlorpromazine pressure alone selected for mutations in *acrR*, a transcriptional repressor of the *acrAB* operon, across all replicates (Fig 8A and S3 File). Most of these were gene disruptions due to IS-element insertions (Fig 8A and S3 File). Loss of AcrR results in overexpression of the AcrAB efflux pump, as well as other determinants of drug resistance [14,48,49]. In trimethoprim and chlorpromazine, *acrR* mutations were seen in 5 of the 6 lineages in addition to mutations in *mgrB* or *pitA* (Fig 8A and S3 File). Thus, unlike genetic inhibition, EPI-induced antibiotic hypersensitivity could be overcome by *E. coli* due to by-pass mutations in the transcriptional regulator *acrR*. Finally, since AcrAB is an efflux pump with a broad substrate range, we expected lineages evolved in chlorpromazine to show enhanced survival in multiple antibiotics. Indeed, evolved populations showed better growth on media supplemented with ciprofloxacin and chloramphenicol,

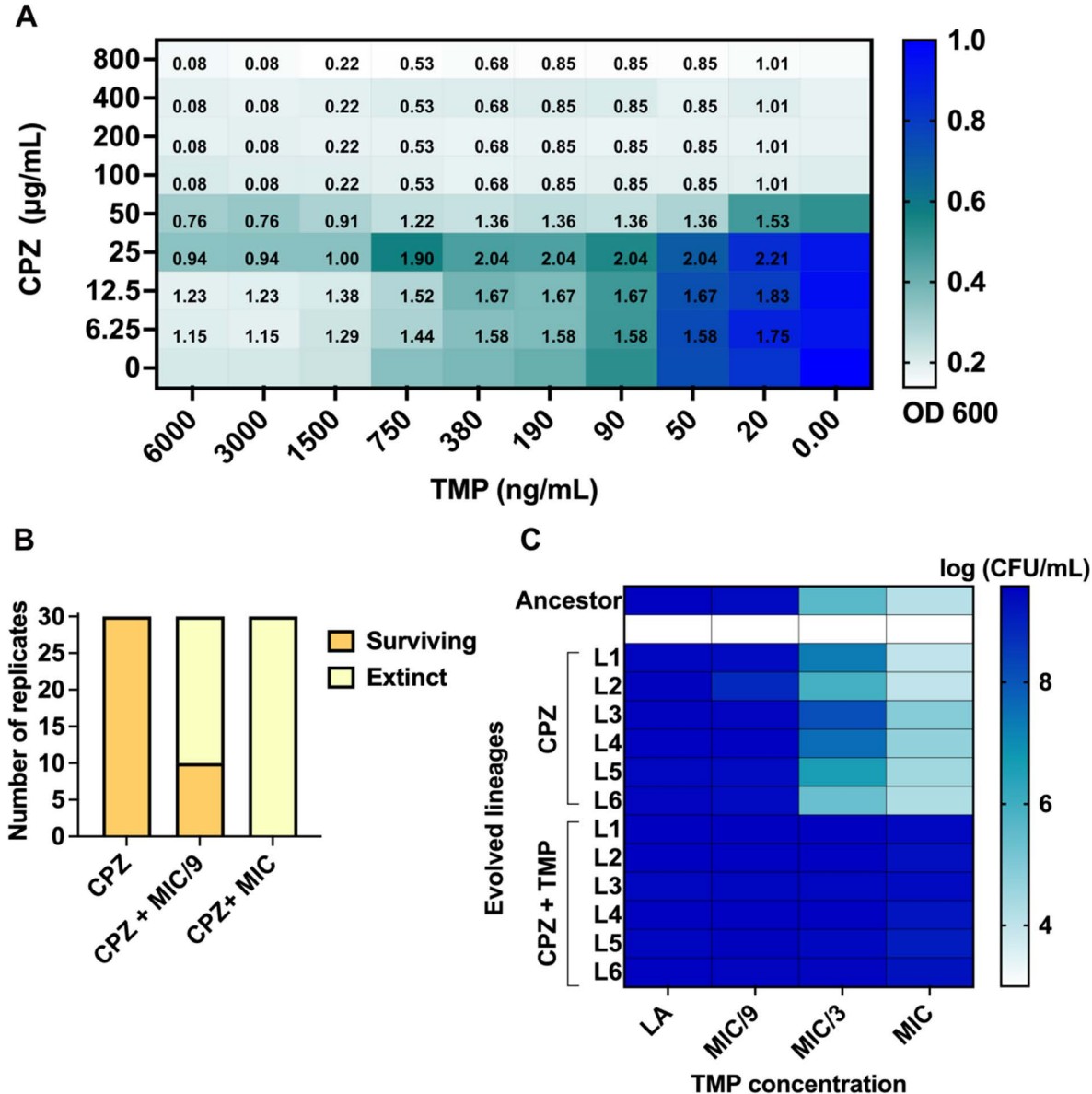

**Fig 7. Pharmacological inhibition of AcrB sensitizes *Escherichia coli* to trimethoprim, but not over evolution in low drug pressures. (A)** Checkerboard assay map showing growth (OD$_{600}$) of wild-type *E. coli* in the presence of varying concentrations of trimethoprim (TMP) and chlorpromazine (CPZ), normalized to growth in the no antibiotic control well (set to 1). Normalized growth is shown as a color gradient. fractional inhibitory concentration (FIC) values of different drug combinations are provided in corresponding wells of the map. Data represent the mean of growth and FIC values from three independent replicates. **(B)** Number of surviving replicate populations of wild-type *E. coli* after 5 cycles of growth in 50 μg/mL chlorpromazine (CPZ) or CPZ and trimethoprim at MIC or MIC/9. **(C)** Colony-forming efficiencies of six lineages (L1–L6) of wild-type *E. coli* evolved in chlorpromazine alone (CPZ) or chlorpromazine and MIC/9 trimethoprim (CPZ+TMP) for 20 growth cycles (~140 generations) on media supplemented with different concentrations of trimethoprim. Data represent the mean of log colony-forming efficiencies from three independent replicates as a color gradient. The data underlying this Figure can be found in S2 File.

both known substrates of the AcrAB-TolC efflux pump (Fig 8B and S2 File). However, no change was seen on media supplemented with colistin, which is not an efflux pump substrate (Fig 8B and S2 File). Thus, at low drug pressure, EPI exposure facilitated evolution of multidrug resistance highlighting a risk of using this strategy for adjuvant design.

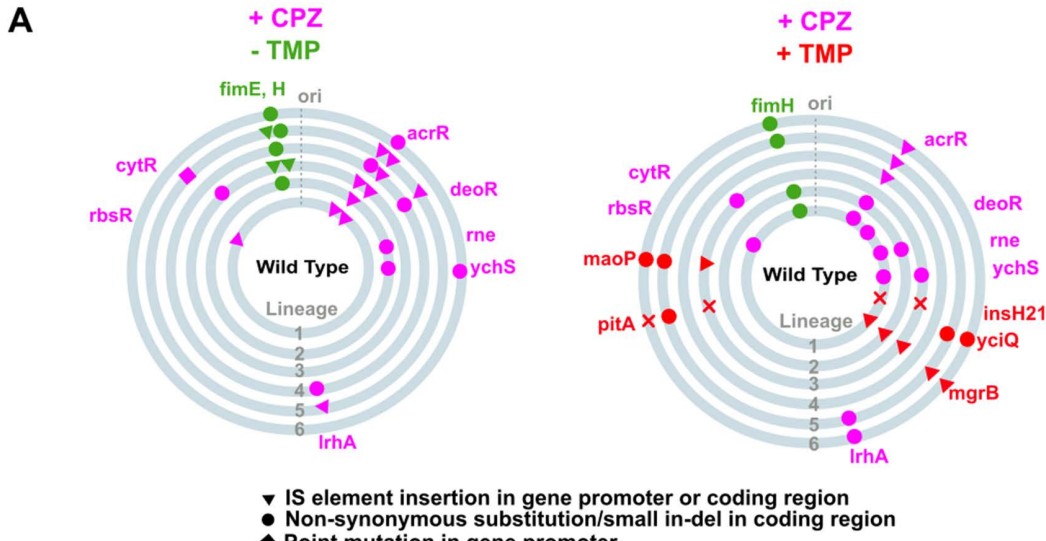

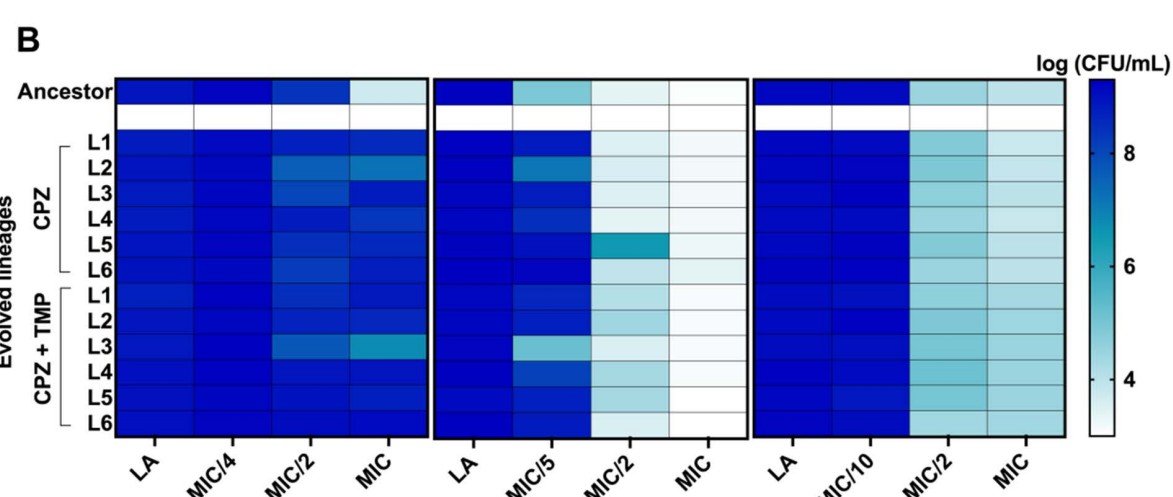

**Fig 8. Adaptation to chlorpromazine and trimethoprim drives multidrug resistance. (A)** Schematic of mutations occurring in six independently evolved lineages of wild-type *E. coli* (L1 to L6) in chlorpromazine (+CPZ −TMP) or chlorpromazine and low trimethoprim (+CPZ+TMP). Each circle represents and independent lineage. Mutations in magenta are those that occur in CPZ alone, and mutations in red occurred in CPZ and TMP. Mutations that were observed in evolution in drug-free media are in green. All listed mutations occurred in at least 20% of the population, were present in 2 or more lineages and/or were in genes involved in folate metabolism. **(B)** Colony-forming efficiencies of evolved lineages of wild-type *E. coli* on media supplemented with varying concentrations of chloramphenicol, ciprofloxacin, and colistin. Data represent the mean of log colony-forming efficiencies from three independent replicates and are represented as a color gradient. The data underlying this Figure can be found in S2 and S3 Files.

## Discussion

The intrinsic resistome of gram-negative bacteria represents a substantial barrier for antibiotics. Attenuating this barrier holds the promise of repurposing or reusing otherwise ineffective antimicrobials for the treatment of gram-negative bacterial infections [3,7,10,11]. A well-known example is the antibiotic vancomycin that inhibits peptidoglycan biosynthesis in

gram-positives but is inactive against gram-negatives due to its poor penetration through the outer membrane. Vancomycin is, however, effective when the cell envelope barrier is breached. Based on this principle, derivatives and formulations of vancomycin have been developed that permeate better into the gram-negative cell and could serve as future antibiotics [50,51]. In the present study, we demonstrate that *E. coli* is rendered hypersensitive to multiple classes of antibiotics upon perturbing pathways that constitute its intrinsic resistome. Interfering with intrinsic resistance mechanisms was able to reverse chromosomally encoded resistance as well as retard *de novo* resistance evolution in high drug selection regimes by ensuring culture sterilization. However, at sub-lethal drug pressures adaptive evolution rapidly overcame antibiotic hypersensitivity. The extent of recovery differed dramatically between hypersensitive mutants and was determined by the ability of antibiotic-specific beneficial mutations to overcome defects in intrinsic resistance pathways. Our analyses showed that perturbing efflux was a more robust strategy for resistance proofing than cell wall biogenesis, even when adaptive mutations did not directly engage with efflux/cell wall pathways. Finally, though targeting efflux genetically or pharmacologically were comparable in short term, they diverged substantially on evolutionary time scales. This was because of rapid evolution in transcriptional regulators of efflux genes that could by-pass the effects of chemical inhibitors. We propose that this divergence poses a serious hurdle to sensitization and resistance-proofing efforts going forward.

In the present study, we comprehensively characterized genes in *E. coli* that modulate susceptibility to trimethoprim and chloramphenicol, two chemically distinct antibiotics with diverse cellular targets and resistance mechanisms. For this purpose, we have used the Keio mutant collection [40] as a source of *E. coli* gene deletion strains. The Keio collection has served as a useful resource for identifying antibiotic-resistant and hypersensitive mutants against multiple antibiotics. In more recent work, this mutant library has been coupled with laboratory evolution providing a powerful tool kit for revealing the distribution of fitness effects of gene deletions and determining how epistasis alters evolvability [29,30,52]. Our study uses much the same approach to understand the genetic determinants of antibiotic hypersensitivity and changes in the fitness landscape of *E. coli* due to perturbations in intrinsic resistance pathways. Indeed, the hits that we focused on here have also been identified for other antibiotics reiterating their antibiotic-agnostic role in modulating drug susceptibility [29,30,52]. The enrichment of cell wall-related genes in the hypersensitivity screen for both antibiotics reinforced the importance of this permeability barrier for gram-negative bacteria. Strikingly, knockouts of genes at different steps of cell wall biogenesis ranging from lipid A (*lpxM*) and core oligosaccharide (*kdsO*) biosynthesis to LPS transport (*bamB*) came up in our screen, likely pointing to multiple opportunities for interventions within this pathway for antibiotic sensitization. Some of these hits, like BamA and LpxC proteins, are also being viewed as potential antibiotic targets themselves [53,54]. Yet, even though the number of cell wall mutants picked up by our screen was large, preventing resistance evolution by inhibiting two independent pathways within cell wall biosynthesis, i.e., *rfaG* and *lpxM*, had limited success. Indeed, this inadequacy is resonant with the findings reported by Maharramov and co-authors [55] showing that the risk of resistance evolution is low only when dual inhibition of membrane integrity and at least one more pathway is imposed on bacteria. A novel result we present here, therefore, is that though an important pathway of intrinsic resistance, taregting cell wall biosynthesis may prove ineffective for preventing de novo resistance evolution.

On the other hand, loss of the major efflux protein AcrB was a more effective strategy for antibiotic sensitization as well as preventing de novo resistance evolution at high and low drug pressure. The AcrAB-TolC complex has long been recognized as the primary antibiotic efflux pathway in Enterobacteriaceae [56]. Studies have demonstrated that mutations in *acrAB* result in more robust reduction in drug MICs than other pumps encoded by the genome of *E. coli* [57,58]. In part, this may be due to the broad substrate repertoire of AcrAB. The polysubstrate specificity of this efflux pump is due to at least 4 different channels within the AcrAB complex that export chemically diverse molecules, including several antibiotics, dyes such as acriflavine and ethidium bromide, or detergents like dodecyl β-D-maltoside [59,60]. Moreover, while other efflux pumps transport molecules across the inner membrane, i.e., from cytosol to the periplasm, the AcrAB-TolC complex is capable of exporting molecules across the outer membrane as well [56]. As a result, many other efflux pumps are dependent on the activity of AcrAB-TolC. Taken together with the results of the present study, these attributes of

AcrAB-TolC strongly support its candidacy as a target for antibiotic sensitization. It is also noteworthy that novel genetic determinants of antibiotic sensitivity were also identified by our genetic screen. For instance, knockout of DNA methylase Dam and cytoskeletal protein RodZ, both sensitized *E. coli* to trimethoprim. In the same vein, knockouts of metabolic enzymes like acetate kinase AckA were identified as determinants of chloramphenicol sensitivity. These new pathways remain to be mechanistically investigated and may provide viable alternatives to cell wall and efflux proteins in the design of resistance breakers.

Our experiments differentiated between two possible models for why impeding intrinsic resistance pathways altered the adaptability of *E. coli* to trimethoprim to different extents. The first model invokes a lower population size, and hence lower mutation supply, for hypersensitive mutants under antibiotic pressure. However, this model is unlikely to be true, as similar population sizes were observed for all three hypersensitive mutants at low and high trimethoprim pressure. An alternative model, which we provide more compelling evidence for, is that adaptive mutations have different benefits across hypersensitive backgrounds. Indeed, compromised efflux through AcrAB dampened the benefit of adaptive mutations to a greater extent than perturbation of cell wall. Notably, targeting efflux for preventing the evolution of high-level resistance against antibiotics like fluoroquinolones has been demonstrated earlier [31–34,61]. However, it is important to emphasize that adaptive mutations against trimethoprim do not directly engage efflux pathways, unlike for fluoroquinolones where resistance-conferring mutations frequently map to efflux pumps or their regulators [31–34,61]. Thus, our study shows that the effects of inhibiting efflux can interfere with mechanisms of resistance even when they do not directly involve antibiotic transporters. On the other hand, the same may not hold true for cell wall perturbation. We argue, therefore, that understanding the relationship between intrinsic drug susceptibility and adaptations that emerge under drug pressure is crucial to predict evolvability of resistance.

Trimethoprim resistance in *E. coli*, which we have used in the present study as a model for adaptive evolution, is driven by a network of mutations at multiple chromosomal loci [37–39,42]. Importantly, though we performed evolution experiments starting with different genetic backgrounds, the mutational landscape of trimethoprim adaptation remained similar across wild-type and hypersensitive strains. This is in contrast to earlier work that identified general, rather than drug-specific, compensatory evolution in AcrAB-deficient bacteria due to transcriptional upregulation of other efflux pumps [62–64]. This disparity possibly represents the different fitness landscapes of adaptive mutations across antibiotic classes. The most frequent adaptive strategy observed by us in the present work was enhanced expression of the target of trimethoprim, i.e., DHFR. This adaptation was driven by an increase in *folA* gene copy number, loss-of-function mutations in *mgrB* or activating mutations in the *folA* gene promoter, all of which we have reported earlier [38,39,42]. Interestingly, we identified two additional loci, i.e., DsbA and PitA, as mediators of trimethoprim resistance in the present study, expanding our understanding of the repertoire of adaptive mutations against this antibiotic in *E. coli*. Mutations in DsbA conferred resistance by activating the PhoQP pathway, similar to MgrB. Curiously, spontaneous *dsbA* mutations, though similar to *mgrB* in their mechanism of trimethoprim resistance, have never been isolated in *E. coli* challenged with the antibiotic. In the present study too, *dsbA* mutations only evolved in an *rfaG* knockout, which is itself slightly slower growing than the wild type. This suggests that hypersensitive strains may use otherwise costly mutations to adapt to antibiotics, possibly due to "diminishing returns epistasis," i.e., selection of high benefit mutations is possible despite large costs when genetic backgrounds have lower fitness to begin with. Notably, inactivation of DsbA is being explored as a strategy for anti-virulence in *Salmonella* [65] and *E. coli* [66], given its role as a periplasmic chaperone. Our results suggest that these approaches may run the risk of collaterally selecting for trimethoprim resistance. On the other hand, loss of PitA is a novel mechanism of resistance, involving neither PhoQP nor DHFR. Earlier work has implicated PitA in resistance to spectinomycin possibly due to a change in proton homeostasis across the inner membrane [67]. It is known that treatment of *E. coli* with trimethoprim drives cytosolic acidification [68], suggesting that loss of PitA may confer resistance by preventing drug-induced proton distribution across the membrane.

Our results have significant implications for the emerging field of "resistance proofing" [28,69,70], i.e., designing antimicrobial treatments that are less amenable to developing resistance. Given rapid rates of resistance evolution in pathogenic bacteria, resistance-proofing new antimicrobials may be essential to prolong their life time and sustain future drug discovery efforts. The recent emergence of bedaquiline resistance in *Mycobacterium tuberculosis* within a few years of its clinical deployment is illustrative of this need [71]. Different ideas are being explored for resistance proofing, such as combining antibiotics, targeting multiple steps within the same metabolic pathway, identifying antibiotics with nonprotein targets, or reducing access to adaptive mutations by inhibiting transcription-coupled repair [70,72]. We propose inhibiting intrinsic resistance pathways as an alternative to combining antibiotics. This strategy has the advantage of exploiting relatively well-worked-out aspects of bacterial physiology, and does not run the risk of antagonism that is frequently observed between antibiotics. Additionally, it builds on a growing body of literature that attributes treatment failures to "general" rather than drug-specific resistance mechanisms in emerging pathogens like *Acinetobacter baumanii* [73]. However, a major hurdle appears to be the evolutionary instability of antibiotic hypersensitivity at sub-lethal drug pressures. Like with antibiotic combinations, therefore, strong stewardship may be needed to ensure that sub-MIC exposures are minimized for this strategy to remain effective.

Finally, though genetic screens have served as a major experimental strategy for discovering novel targets for antibacterial, the lack of consistency in evolutionary outcomes between genetically and pharmacologically sensitized bacteria highlights a major shortcoming of this approach. Much effort has been directed toward the design of EPIs over the past decade, given the critical role of AcrAB in drug efflux [11,74–77]. While this approach is likely to be effective for sensitizing drug-resistant bacterial strains, rapid evolution of EPI resistance due to mutations in transcriptional regulators such as AcrR represents a substantial hurdle. Possible solutions include designing higher-affinity inhibitors whose efficacy is not significantly affected by an increase in expression level of the pump. Alternatively, other components of this complex such as TolC may be more promising targets, though druggability and pathways of resistance remain to be investigated.

## Materials and methods

### Bacterial strains and culture conditions

*E. coli* strains were cultured in Luria-Bertani (LB) broth or grown as colonies on LB agar supplemented with antibiotics when required. Broth cultures were grown at 37 °C with shaking at 180 rpm, and plates were incubated at 37 °C. Frozen stocks were prepared by mixing sterile 50% glycerol and saturated cultures in a 1:1 ratio, and stored at −80 °C until further use.

The strains used in this study are shown in Table 1.

### Genetic screen for antibiotic hypersensitivity in *E. coli*

We screened the Keio knockout collection [40], which contains single gene knockouts of all nonessential genes in *E. coli*, to identify genetic determinants of antibiotic hypersensitivity. For this screen, 150 µL of LB media was added to the wells of a 96-well microtitre plates in three sets. Trimethoprim (300 ng/mL) or chloramphenicol (3 µg/mL) were added to sets 1 and 2, respectively, while no antibiotic was added to set 3. These concentrations of antibiotics were selected since they resulted in ~50% growth inhibition for wild-type *E. coli.* Cultures of *E. coli* K-12 BW25113 and all knockouts from the Keio collection were grown overnight in drug-free LB and inoculated (1%) into the microtitre plates. Plate cultures were then incubated for 18–20 h at 37 °C without agitation. Growth was measured as optical density at 600 nm ($OD_{600}$) using a multi-mode plate reader (Revvity, USA). Growth ($OD_{600}$) of knockout strains was normalized to the wild type (set to 1) in respective growth media. For each mutant, two biological replicate measurements were recorded. Normalized growth values for the entire knockout collection were then plotted and fit to a Gaussian distribution. Knockouts were classified as hypersensitive if they grew less than 2 standard deviations from the median of the distribution. Knockouts that also showed poor growth in drug-free LB were then manually removed.

**Table 1. List of bacterial strains used in the study.**

| S.No. | Strain/s | Description | Source/Reference |
|---|---|---|---|
| 1. | Keio single gene knockout collection | All single gene knockouts of nonessential genes in *E. coli* K-12 BW25113 background [40] | National BioResource Project, National Institute of Genetics, Japan |
| *E. coli* reference strains | | | |
| 1. | *E. coli* K-12 MG1655 | *E. coli* WT, used as reference strain for phenotypic and genotypic comparisons | Available in the laboratory |
| 2. | *E. coli* K-12 BW25113 | *E. coli* WT, used as reference strain for genetic screen | Available in the laboratory |
| Gene knockouts in *E. coli* MG1655 | | | |
| 1. | *E. coli* ΔrfaG | Isogenic knockout of *rfaG* gene (Δ*rfaG::Kan*) | Generated by P1 transduction using the corresponding knockout strain from the Keio Collection [40] as the allele donor |
| 2. | *E. coli* ΔacrB | Isogenic knockout of *acrB* gene (Δ*acrB::Kan*) | Generated by P1 transduction using the corresponding knockout strain from the Keio Collection [40] as the allele donor |
| 3. | *E. coli* ΔnudB | Isogenic knockout of *nudB* gene (Δ*nudB::Kan*) | Generated by P1 transduction using the corresponding knockout strain from the Keio Collection [40] as the allele donor |
| 4. | *E. coli* ΔlpxM | Isogenic knockout of *lpxM* gene (Δ*lpxM::Kan*) | Generated by P1 transduction using the corresponding knockout strain from the Keio Collection [40] as the allele donor |
| 5. | *E. coli* ΔdsbA | Isogenic knockout of *dsbA* gene (Δ*dsbA::Kan*) | Generated by P1 transduction using the corresponding knockout strain from the Keio Collection [40] as the allele donor |
| 6. | *E. coli* ΔpitA | Isogenic knockout of *pitA* gene (Δ*pitA::Kan*) | Generated by P1 transduction using the corresponding knockout strain from the Keio Collection [40] as the allele donor |
| 7. | *E. coli* ΔmgrB | Isogenic knockout of *mgrB* gene (Δ*mgrB::Kan*) | [38] |
| 8. | *E. coli* ΔrfaGΔdsbA | Double knockout of *rfaG* (Δ*rfaG::FRT*) and *dsbA* (Δ*dsbA::Kan*) genes | Generated by P1 transduction using the corresponding knockout strain from the Keio Collection [40] as the allele donor |
| 9. | *E. coli* ΔrfaGΔpitA | Double knockout of *rfaG* (Δ*rfaG::FRT*) and *pitA* (Δ*pitA::Kan*) genes | Generated by P1 transduction using the corresponding knockout strain from the Keio Collection [40] as the allele donor |
| 10. | *E. coli* ΔrfaGΔmgrB | Double knockout of *rfaG* (Δ*rfaG::FRT*) and *mgrB* (Δ*mgrB::Kan*) genes | Generated by P1 transduction using the corresponding knockout strain from the Keio Collection [40] as the allele donor |
| Isolates derived from adaptive laboratory evolution used for phenotypic analysis | | | |
| 1. | *E. coli* TMP$^R$ — 1TR1 | Trimethoprim-resistant isolate derived from long-term evolution harboring a *folA* $C_{-35}T$ mutation. | [39] |
| 2. | *E. coli* TMP$^R$ — 1TR1 ΔrfaG | Isogenic knockout of *rfaG* gene (Δ*rfaG::Kan*) in trimethoprim-resistant isolate *E. coli* TMP$^R$ — 1TR1 | Generated by P1 transduction using the corresponding knockout strain from the Keio Collection [40] as the allele donor |
| 3. | *E. coli* TMP$^R$ — 1TR1 ΔacrB | Isogenic knockout of *acrB* gene (Δ*acrB::Kan*) in trimethoprim-resistant isolate *E. coli* TMP$^R$ — 1TR1 | Generated by P1 transduction using the corresponding knockout strain from the Keio Collection [40] as the allele donor |
| 4. | *E. coli* TMP$^R$ — 1TR1 ΔlpxM | Isogenic knockout of lpxM gene (ΔlpxM::Kan) in trimethoprim-resistant isolate *E. coli* TMP$^R$ — 1TR1 | Generated by P1 transduction using the corresponding knockout strain from the Keio Collection [40] as the allele donor |
| 5. | *E. coli* TMP$^R$ — 2TR1 | Trimethoprim-resistant isolate derived from long-term evolution harboring a *folA* — $Trp_{30}Arg$ mutation. | [39] |

*(Continued)*

**Table 1.** (Continued)

| S.No. | Strain/s | Description | Source/Reference |
|---|---|---|---|
| 6. | *E. coli* TMP$^R$ — 2TR1 Δ*rfaG* | Isogenic knockout of *rfaG* gene *(ΔrfaG::Kan)* in trimethoprim resistant isolate *E. coli* TMP$^R$ — 2TR1 | Generated by P1 transduction using the corresponding knockout strain from the Keio Collection [40] as the allele donor |
| 7. | *E. coli* TMP$^R$ — 2TR1 Δ*acrB* | Isogenic knockout of *acrB* gene *(ΔacrB::Kan)* in trimethoprim-resistant isolate *E. coli* TMP$^R$ — 2TR1 | Generated by P1 transduction using the corresponding knockout strain from the Keio Collection [40] as the allele donor |
| 8. | *E. coli* TMP$^R$ — 3TR1 | Trimethoprim-resistant isolate derived from long-term evolution harboring a *folA* — Pro$_{21}$Gln mutation. | [39] |
| 9. | *E. coli* TMP$^R$ — 3TR1 Δ*rfaG* | Isogenic knockout of *rfaG* gene *(ΔrfaG::Kan)* in trimethoprim-resistant isolate *E. coli* TMP$^R$ — 3TR1 | Generated by P1 transduction using the corresponding knockout strain from the Keio Collection [40] as the allele donor |
| 10. | *E. coli* TMP$^R$ — 3TR1 Δ*acrB* | Isogenic knockout of *acrB* gene *(ΔacrB::Kan)* in trimethoprim-resistant isolate *E. coli* TMP$^R$ — 3TR1 | Generated by P1 transduction using the corresponding knockout strain from the Keio Collection [40] as the allele donor |
| 11. | *E. coli* TMP$^R$ — 4TR1 | Trimethoprim-resistant isolate derived from long-term evolution harboring *folA* C$_{-35}$T and *mgrB*:Δ1bp mutation. | [39] |
| 12. | *E. coli* TMP$^R$ — 4TR1 Δ*rfaG* | Isogenic knockout of *rfaG* gene *(ΔrfaG::Kan)* in trimethoprim-resistant isolate *E. coli* TMP$^R$ — 4TR1 | Generated by P1 transduction using the corresponding knockout strain from the Keio Collection [40] as the allele donor |
| 13. | *E. coli* TMP$^R$ — 4TR1 Δ*acrB* | Isogenic knockout of *acrB* gene *(ΔacrB::Kan)* in trimethoprim-resistant isolate *E. coli* TMP$^R$ — 4TR1 | Generated by P1 transduction using the corresponding knockout strain from the Keio Collection [40] as the allele donor |
| 14. | *E. coli* ERY$^R$ | Erythromycin-resistant isolate derived from short-term evolution harboring an *rplD* −Lys$_{63}$Gln mutation. | [42] |
| 15. | *E. coli* ERY$^R$ Δ*rfaG* | Isogenic knockout of *rfaG* gene *(ΔrfaG::Kan)* in erythromycin-resistant isolate *E. coli* ERY$^R$ | Generated by P1 transduction using the corresponding knockout strain from the Keio Collection [40] as the allele donor |
| 16. | *E. coli* ERY$^R$ Δ*acrB* | Isogenic knockout of *acrB* gene *(ΔacrB::Kan)* in erythromycin-resistant isolate *E. coli* ERY$^R$ | Generated by P1 transduction using the corresponding knockout strain from the Keio Collection [40] as the allele donor |
| 17. | *E. coli* ERY$^R$ Δ*lpxM* | Isogenic knockout of *lpxM* gene *(ΔlpxM::Kan)* in erythromycin-resistant isolate *E. coli* ERY$^R$ | Generated by P1 transduction using the corresponding knockout strain from the Keio Collection [40] as the allele donor |
| 18. | *E. coli* NAL$^R$ | Nalidixic acid-resistant isolate derived from short-term evolution harboring a *gyrA* −D82G mutation. | [42] |
| 19. | *E. coli* NAL$^R$ Δ*rfaG* | Isogenic knockout of *rfaG* gene *(ΔrfaG::Kan)* in nalidixic acid-resistant isolate *E. coli* NAL$^R$ | Generated by P1 transduction using the corresponding knockout strain from the Keio Collection [40] as the allele donor |
| 20. | *E. coli* NAL$^R$ Δ*acrB* | Isogenic knockout of *acrB* gene *(ΔacrB::Kan)* in nalidixic acid-resistant isolate *E. coli* NAL$^R$ | Generated by P1 transduction using the corresponding knockout strain from the Keio Collection [40] as the allele donor |
| 21. | *E. coli* NAL$^R$ Δ*lpxM* | Isogenic knockout of *lpxM* gene *(ΔlpxM::Kan)* in nalidixic acid-resistant isolate *E. coli* NAL$^R$ | Generated by P1 transduction using the corresponding knockout strain from the Keio Collection [40] as the allele donor |
| 22. | *E. coli* RFP$^R$ | Rifampicin-resistant isolate derived from short-term evolution harboring an *rpoB* −S512P mutation. | [42] |
| 23. | *E. coli* RFP$^R$ Δ*rfaG* | Isogenic knockout of *rfaG* gene *(ΔrfaG::Kan)* in rifampicin-resistant isolate *E. coli* RFP$^R$ | Generated by P1 transduction using the corresponding knockout strain from the Keio Collection [40] as the allele donor |
| 24. | *E. coli* RFP$^R$ Δ*acrB* | Isogenic knockout of *acrB* gene *(ΔacrB::Kan)* in rifampicin-resistant isolate *E. coli* RFP$^R$ | Generated by P1 transduction using the corresponding knockout strain from the Keio Collection [40] as the allele donor |

*(Continued)*

**Table 1.** (Continued)

| S.No. | Strain/s | Description | Source/Reference |
|---|---|---|---|
| 25. | *E. coli* RFP$^R$ Δ*lpxM* | Isogenic knockout of *lpxM* gene (Δ*lpxM::Kan*) in rifampicin-resistant isolate *E. coli* RFP$^R$ | Generated by P1 transduction using the corresponding knockout strain from the Keio Collection [40] as the allele donor |
| 26. | *E. coli* AMX$^R$ | Amoxicillin-resistant isolate derived from short-term evolution harboring an *ampC* −G-25T mutation. | [42] |
| 27. | *E. coli* AMX$^R$ Δ*rfaG* | Isogenic knockout of *rfaG* gene (Δ*rfaG::Kan*) in amoxicillin-resistant isolate *E. coli* AMX$^R$ | Generated by P1 transduction using the corresponding knockout strain from the Keio Collection [40] as the allele donor |
| 28. | *E. coli* AMX$^R$ Δ*acrB* | Isogenic knockout of *acrB* gene (Δ*acrB::Kan*) in amoxicillin-resistant isolate *E. coli* AMX$^R$ | Generated by P1 transduction using the corresponding knockout strain from the Keio Collection [40] as the allele donor |
| 29. | *E. coli* AMX$^R$ Δ*lpxM* | Isogenic knockout of *lpxM* gene (Δ*lpxM::Kan*) in amoxicillin-resistant isolate *E. coli* AMX$^R$ | Generated by P1 transduction using the corresponding knockout strain from the Keio Collection [40] as the allele donor |

## Measuring antibiotic resistance level

**Colony formation on drug-supplemented solid media.** Serial dilution spot assays were employed to check for antibiotic sensitivities of wild-type and knockout strains of *E. coli* to different antibiotics. Cultures of appropriate strains were grown overnight in drug-free LB, and a 10-fold serial dilution was set up. Diluted cultures (10 μL) were immediately spotted on LA alone, or LA supplemented with different concentrations of the required antibiotics. Plates were incubated at 37 °C overnight for 15–18 h and colony-forming units/mL (CFUs/mL) were estimated from the highest dilution showing discrete colonies. Log CFU/mL were then compared across strains.

**Broth dilution assays.** The IC50 (concentrations required for 50% growth inhibition) of trimethoprim and other antibiotics for *E. coli* strains was measured using broth dilution assays in 96-well microtitre plates as described earlier [38,39,78]. Briefly, primary cultures of *E. coli* were grown for 6–8 h at 37 °C, and then inoculated (1%) into 150 μL of LB broth in which a 2-fold dilution series of the antibiotic was set up. After 16–18 h of growth at 37 °C OD$_{600}$ was measured. Growth for each strain was normalized to growth in the absence of the antibiotic (set to 1). The data were then fitted to a nonlinear regression log inhibitor versus response curve using Graphpad Prism (version 9.1.4). Following equation was used to determine IC50 value.

$$Y = Min + (Max - Min)/1 + 10((Log(IC50)-X) * Hill\ Slope))$$

where Min, Minimum OD value; Max, Maximum OD value (=1); and Hill Slope, Slope of curve after fitting.

**Generation of knockouts in *E. coli* K-12 MG1655.** All knockouts in the study were generated using P1 transduction by moving the kanamycin resistance-marked gene deletion from the donor Keio strain to the appropriate recipient strain of *E. coli.* Knockouts were confirmed by using polymerase chain reactions with gene-specific primers or a combination of primers from the gene and the kanamycin resistance cassette. The kanamycin resistance cassette was excised using plasmid pCP20 plasmid-expressing FLP recombinase [79]. Double knockouts were generated by first excising the kanamycin resistance cassette from a single gene knockout, and then introducing the second kanamycin-marked knockout by P1 transduction.

**Adaptive laboratory evolution of trimethoprim resistance.** The methodology followed for laboratory evolution of trimethoprim resistance has been described in Vinchhi and colleagues [39]. Briefly, wild-type and appropriate knockout strains were cultured either in 150 μL of LB alone, or LB supplemented with 100 ng/mL trimethoprim, 50 μg/mL of chlorpromazine or both in 96-well plates. For each strain and media combination, 30 replicates were set up for

short-term evolution (i.e., 5 serial transfers), of which 6 were taken forward for longer duration of evolution (i.e., 20 serial transfers, ~140 generations). Plates were incubated at 37 °C, and 1% of the culture was passaged into fresh media every 24 h. Aliquots were frozen every 5 passages and stored at −80 for all evolving strains. Trimethoprim resistance in evolving lineages was determined by reviving the frozen stocks in drug-free media and determining CFU/mL at three different concentrations of trimethoprim, i.e., MIC, MIC/3, and MIC/9 of the wild type ancestor.

**Genome sequencing of laboratory-evolved trimethoprim-resistant populations.** Genome sequencing of the evolved population was performed as described in Vinchhi and Yelpure and colleagues and Vinchhi and colleagues [39,78]. Briefly, frozen stocks of evolved populations from the final time point were revived in drug-free media overnight. Genomic DNA was extracted from these cultures using the phenol:chloroform:isoamyl alcohol method. Paired-end whole-genome Next Generation sequencing was then performed on a MiSeq system (Illumina, USA) with read lengths of 150−200 bps. Library preparation and sequencing services were provided by Eurofins, India and Genotypic Technology Pvt. The genome of *E. coli* K-12 MG1655 (Genbank:U00096.3) was used as reference. Raw sequencing data were analyzed using Breseq in population mode using default parameters [80,81] and all point mutations, structural mutants and transposition events were identified. Next, mutations that were present in the ancestor strain and present at lower than 20% frequency were excluded. The remaining mutations were taken forward for further analyses. Raw sequencing reads are available on GenBank (PRJNA1266351).

**Quantification of phoP promoter activity.** The reporter plasmid pUA66-$P_{phoP}$-GFP, containing the *E. coli* MG1655 *phoP* gene promoter fused to a GFP reporter was obtained from Dharmacon/Revvity (catalogue number ID: PEC3876-98154831) [82] and transformed into the appropriate strains of *E. coli*. Cultures of reporter plasmid-harboring strains were grown overnight and then diluted 1:5 in fresh LB medium. For diluted cultures, bacterial density ($OD_{600}$) and GFP fluorescence (Ex: 485 nm, Em: 519 nm) was measured. PhoP promoter activity was assessed by normalizing GFP fluorescence to $OD_{600}$. Normalized promoter activity is reported as fold changes relative to the wild-type strain (set to 1).

**Immunoblotting for DHFR expression.** Immunoblotting was used to estimate DHFR protein levels in wild-type and knockout strains of *E. coli.* Anti-DHFR polyclonal IgG was used to estimate DHFR protein levels as described earlier [39,42,83,84]. Briefly, exponential phase cultures of wild-type and knockout strains of *E. coli* were lysed, 5 µg of total protein was loaded on a 15% SDS-PAGE gel, and electroblotted on a PVDF membrane. After blocking with 5% BSA, the membrane was probed with 100 ng/mL of anti-DHFR polyclonal IgG, followed by HRP-linked anti-rabbit IgG (1:10,000). Bands were detected using chemiluminescence. FtsZ protein levels were also estimated as a loading control.

**Checkerboard assay for trimethoprim and chlorpromazine.** Inhibitory effects of different combinations of trimethoprim and chlorpromazine on wild-type *E. coli* were measured using a checkerboard assay. In a 96-well microtitre plate, a 2-fold serial dilution series of trimethoprim (vertical), and 2-fold serial dilution series of chlorpromazine (horizontal) were set up to give a final volume of 150 µL in each well. Cultures of wild-type *E. coli* were grown for 6–8 h at 37 °C, and then inoculated (1%) into each well. Optical densities were measured at 600 nm after 16–18 h of incubation at 37 °C. $OD_{600}$ values were normalized to growth in the absence of the antibiotic (set to 1).

Drug interactions were analyzed by calculating fractional inhibitory concentration (FIC) indices for each combination [85], using the following formula

$$FIC = MIC_{(AB)}/MIC_{(A)} + MIC_{(BA)}/MIC_{(B)}$$

where $MIC_{(AB)}$ is the MIC of drug A in the presence of drug B, $MIC_{(A)}$ is the MIC for drug A alone, $MIC_{(BA)}$ is the MIC of drug B in the presence of drug A, and $MIC_{(B)}$ is the MIC for drug B alone.

## Supporting information

**S1 File. Results of antibiotic hypersensitivity screen and short-listed hypersensitive knockouts.**
(XLSX)

**S2 File. Colony-forming efficiencies and IC50 values of genetically manipulated/evolved strains against trimethoprim and other antibiotics.**
(XLSX)

**S3 File. Lists of mutations identified by genome sequencing of trimethoprim and chlorpromazine evolved *Escherichia coli* populations.**
(XLSX)

**S1 Text. Raw images.**
(PDF)

## Author contributions

**Conceptualization:** Manasvi Balachandran, Nishad Matange.

**Data curation:** Nishad Matange.

**Formal analysis:** Manasvi Balachandran, Nishad Matange.

**Funding acquisition:** Nishad Matange.

**Investigation:** Manasvi Balachandran, Rohini Chatterjee, Ishaan Chaudhary, Chinmaya Jena, Nishad Matange.

**Methodology:** Manasvi Balachandran, Rohini Chatterjee.

**Project administration:** Nishad Matange.

**Resources:** Nishad Matange.

**Supervision:** Nishad Matange.

**Visualization:** Manasvi Balachandran, Nishad Matange.

**Writing – original draft:** Nishad Matange.

**Writing – review & editing:** Manasvi Balachandran, Nishad Matange.

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
