## [Editor Report · Decision Letter 0]

30 May 2025

Dear Dr Matange,

Thank you for submitting your manuscript entitled "Antibiotic sensitisation and resistance proofing by impeding pathways of intrinsic resistance in Escherichia coli" for consideration as a Research Article by PLOS Biology.

Your manuscript has now been evaluated by the PLOS Biology editorial staff, as well as by an academic editor with relevant expertise, and I am writing to let you know that we would like to send your submission out for external peer review.

Once your full submission is complete, your paper will undergo a series of checks in preparation for peer review. After your manuscript has passed the checks it will be sent out for review. To provide the metadata for your submission, please Login to Editorial Manager (https://www.editorialmanager.com/pbiology) within two working days, i.e. by Jun 01 2025 11:59PM.

Kind regards,

Melissa

Melissa Vazquez Hernandez, Ph.D.

Associate Editor

PLOS Biology

---

## [Decision Letter · Decision Letter 1]

15 Jul 2025

Dear Dr Matange,

Thank you for your patience while your manuscript "Antibiotic sensitisation and resistance proofing by impeding pathways of intrinsic resistance in Escherichia coli" was peer-reviewed at PLOS Biology. It has now been evaluated by the PLOS Biology editors, an Academic Editor with relevant expertise, and by three independent reviewers. My sincere apologies for the delay.

In light of the reviews, which you will find at the end of this email, we would like to invite you to revise the work to thoroughly address the reviewers' reports. As you will see below, there is a general concern regarding missing literature, and the novelty not being clear. Reviewer 1 is suggests an additional experiment to test the ability of the WT to 2X/4X MIC and use it as control for the rescue experiments. The reviewer also mentions missing literature that the authors should cite. Reviewer 2 gives suggestions to make the text an "easier" read and streamline it. Reviewer 3 says that the conceptual novelty of the study is not clear, and misses some quantitative rigor. We agree with all reviewer concerns and would require some additional experimental revisions to address them, as we consider that this would strengthen the work.

IMPORTANT: after discussion with the Academic Editor and the reviewers, we would like to emphasize that you should explain better the novelty in light of the several related studies mentioned by the reviewers. Additionally, we do request that you do the control experiment suggested by Reviewer 1 as it will help to understand the lower evolvability of the knockouts in terms of survival versus mutation access.

Given the extent of revision needed, we cannot make a decision about publication until we have seen the revised manuscript and your response to the reviewers' comments. Your revised manuscript is likely to be sent for further evaluation by all or a subset of the reviewers.

**IMPORTANT - SUBMITTING YOUR REVISION**

*Re-submission Checklist*

*Published Peer Review*

*PLOS Data Policy*

*Blot and Gel Data Policy*

Sincerely,

Melissa

Melissa Vazquez Hernandez, Ph.D.

Associate Editor

PLOS Biology

REVIEWERS' COMMENTS:

Reviewer #1:

The aim of this experimental study is to assess the potential of limiting antibiotic resistance emergence by reducing the intrinsic resistance of bacteria. Using E. coli as a model bacterium, the authors identify central genes in the intrinsic resistance to chloramphenicol and trimethoprim (lpxM, rfaG, acrB). They then demonstrate that deletion (as a proxy for inhibition) of these genes indeed increases sensitivity and reduces the ability to evolve resistance to regular doses of trimethoprim. However, at reduced doses of trimethoprim, resistance emerges by adaptive evolution in all three cases. The acrB efflux mutant appears to be more limited compared to the others and thus the authors test the ability of chemical inhibitors of efflux to similarly suppress resistance emergence. Importantly they find that bacterial adaptation rapidly overcomes the chemical inhibitor, identifying a significant limitation of this approach for treatment.

There is little one can criticize in the conducted work. The authors clearly know what they are talking about and all experiments include relevant controls and analyses. The paper provides enough details of its methodology so that its experiments could be reproduced. The results are well presented and the manuscript is well written (clear and accessible to a wide audience). The main claims are:

1) There is a significant overlap in the intrinsic resistance to trimethoprim and chloramphenicol, in which lpxM, rfaG and acrB are central players.

2) Reduction of intrinsic resistance by deletion or inhibition reduces resistance emergence

3) Inhibition of intrinsic resistance is rapidly overcome by evolution.

Generally, these points are each well supported by the presented data. The findings would be a significant contribution and suitable for publication with PLOS Biology. However, one aspect of claim 2 may be strengthened (although this is not required; major point 1). I also have some concerns about fair treatment of the literature (major point 2).

Major points

MP1.

One aspect is that the rescue experiments (Fig 4a) are done at MIC of the wild type. As the three mutants are hypersensitive, the used dosage is effectively 4xMIC (acrB, lpxM) or 2xMIC (rfaG). What is not fully clear is whether the observed extinction is caused by genetic limitation of the mutant (inaccessibility of genetic pathways), or whether the rescaling of the susceptibility is what causes the constraint (greater step required). A control experiment here would be to test the ability of the WT to adapt to 2xMIC and 4xMIC. As only 3/6 WT lineages survived the MIC challenge, a high number of replicate cultures would be required to achieve the required resolution with this analysis.

MP2.

The authors somewhat neglect a large body of work on resistance evolvability. Here, there is considerable recent work (e.g. https://pubmed.ncbi.nlm.nih.gov/39984459/
https://pubmed.ncbi.nlm.nih.gov/32561723/ ). I am especially concerned that the authors do not relate their work to two previous pieces of Guillaume Cheverau and Tobias Bollenbach where the Keio library was used to identify genetic constraints on evolvability, importantly including chloramphenicol, trimethoprim and their combination (https://pubmed.ncbi.nlm.nih.gov/26581035/
https://pubmed.ncbi.nlm.nih.gov/25924924/ ). Within these previous studies genes lpxM and rfaG are indeed already identified and associated to the efficacy of chloramphenicol and trimethoprim. I would strongly encourage the authors to relate their findings to this previous work.

Minor points:

L. 55. What are LMICs? Please avoid unexplained abbreviations

L. 80. No references provided for this statement. Some would say that the drug discovery pipeline has been accelerating in the last 10 years.

L. 86. Please rationalize, why chloramphenicol and trimethoprim were chosen as antibiotics of study

L. 123 ff. I understand why the authors chose not to continue the work with the mutants that had generally impaired growth, however, from a treatment perspective, wouldn't these be the most promising candidates?

Reviewer #2 (Markus Seeger):

This nicely presented paper of the Matange lab describes a large set of experiments geared towards understanding the involvement of the drug efflux pump AcrB (among other intrinsic resistance mechanisms) to influence/drive the evolution of drug resistant strains. The study focussed on the two antibiotics trimethoprim and chloramphenicol. The E. coli K-12 lab strain served as model system. In a first set of the study, the authors screened the Keio collection for gene deletions that render E. coli hypersensitive towards trimethoprim or chloramphenicol, thereby assessing the "intrinsic resistome" of E. coli towards these two drugs. Thereby, the authors found around 20 genes, and then focussed mainly on drug efflux pump acrB, the lipopolysaccharide glucosyl transferase I rfaG and Lipid A myristoyl transferase lpxM. Using clean E. coli ΔacrB, ΔrfaG, ΔlpxM strains, the authors then addressed the question how trimethoprim and chloramphenicol treatment drive the evolution of drug resistance in these strains versus the isogenic wt E. coli strain at different drug concentrations. While at a drug concentration of 1xMIC of wt E. coli, not surprisingly none of the mutant strains survived. Nevertheless, the experiments showed that if one can efficiently interfere with these intrinsic resistance pathways, one may achieve desirable "drug-proofing" effects, meaning that one can potentially avoid or at least reduce the emergence of novel mutations that confer drug resistance.

Interestingly, at sub-MIC levels the evolutionary trajectories varied greatly. In the course of these experiments, the authors uncovered novel factors contributing to trimethoprim resistance (i.e. DsbA and PitA). They worked out the mechanisms, i.e. that genetic deletion of dsbA and pitA results in higher expression level of the trimethoprim target DHFR.

In a final set of experiments, the authors compared the evolutionary trajectory of the genetic deletion mutant of acrB versus a pharmacological intervention of drug efflux by including drug efflux inhibitor (namely chlorpromazine). Very interestingly, the authors found that in the presence of chlorpromazine, the most frequent evolutionary consequence was deletion of the acrAB operon repressor acrR, thereby leading to constitutive overproduction of AcrAB and consequently cross-resistance towards a larger set of drugs which are AcrB substrates.

Overall, this is a creative paper that is very nicely written and interesting to read. It contains a large amount of high quality data, which are the result of intense and focussed research, likely over many years. I find many of the interpretations as highly relevant for the field of drug efflux, and the study elaborately leads through the labyrinth of co-acting resistance mechanisms and epistasis trajectories. As a somewhat inevitable side effect, the complexity of the data and their interpretation is rather high; hence it is not an easy read.

The comments below are mostly suggestions how the story may be (further) streamlined to bring the main messages across in a clearer fashion.

Main points

1) I consider the authors' finding of acrR mutations as a result of efflux inhibitor use at sub-MIC concentration as the most striking finding with the highest impact on the drug efflux field. Although the EPI is mentioned in the abstract, the authors can do a better job in saying that there is a fundamental difference in genetically deleting acrB versus inhibiting it pharmacologically with an EPI in terms of evolutionary outcome and that the use of EPIs may lead to undesirable collateral damage, at least in this specific case. Maybe, the authors find a way to stress this already in the title instead of mentioning the "resistance proofing" there, which in my view is less interesting/relevant in the context of this study. Having said all this, it is entirely up to the authors to follow this suggestion.

2) The paper claims in the introduction (line 84) that "targeting intrinsic resistance mechanisms as resistance proofing strategies, i.e. preventing de novo resistance evolution, remains to be explored". While I agree that comparatively little is known, a quick literature search using the terms "evolution drug efflux inhibiton" revealed quite a number of interesting, recent papers: PMID: 36400010, PMID: 32769975, PMID: 39982069. These three in my view relevant papers have not been cited. With a somewhat deeper search, I am sure the authors will find another few good papers in the realm of drug efflux and evolution. I suggest the authors extend their literature search and discuss their findings in light of the newest literature. By no means, these papers undercut the novelty of the authors' work; rather to the contrary, they illustrate that investigation of drug efflux pumps in the context of antimicrobial resistance is a "hot" science topic with a good momentum.

Minor points

1) Line 111 and Fig. 1B: it was interesting to see that some gene deletions of the Keio collection also conferred stronger growth in the presence of antibiotics. What is going on with these?

2) Line 115: mention here what kind of pipeline/method was used to group genes into functional clusters. It is surely mentioned in the methods, but I would quickly state it here as well.

3) Line 130: I was very surprised the authors did not pick up the other components of the AcrAB-TolC system in their Keio screen: acrA and tolC. Any explanation for this?

4) Line 137: it would be of tremendous help for the reader to extend the description of rfaG and lpxM here. Pls consider adding a supplementary Figure drawing the respective pathways and explain in more detail about the physiology of these pathways and how they are potentially linked to trimethroprim and chloramphenicol resistance. It may also not harm to include a drawing of the AcrAB-TolC system in such an additional Supplementary Figure.

5) Line 141: the authors re-introduced the gene deletions into the MG1655 strain. Why didn't the authors use the Keio strains right away?

6) Line 169: suggested to cite papers wherein the term "resistance proofing" has been coined/first used

7) Line 177 and line 192: I do not consider it as surprising that these deletion strains stopped growing considering that the MIC of trimethroprim was several fold above their MIC (= 1xMIC for wt E. coli).

8) Line 251: Consider giving a new section title here. The entire section is quite long.

9) Line 281: Citation 19 does not show a structure of AcrB with chloropramizine, but only a docking result. That's not a proof for specific binding. I would assume that chloropramizine quite broadly inhibits all kinds of pumps. You may also have a brief look at this review on drug efflux inhibition in TB: PMID: 34637511. But quite clearly, chloropramizine also appears to block AcrB function; hence in terms of interpretation of results, all is fine.

10) Line 328: I consider this finding as the core finding of the paper. Pls also include that more clearly into title/abstract such the message is really seen.

11) Line 398-401: I cannot follow the logics of the argument. Pls try to express yourself clearer.

12) Fig. 1C: I would list the "hypotheticals" last.

13) Fig. 7A: The Checkerboard results look a bit odd/non-uniform. Has the experiment been repeated several times? Nevertheless, it is not critical for the evolution experiments performed with chloropramizine.

Reviewer #3:

Balachandran et al. began this study with a screen of the well-characterized Keio single gene deletion collection to identify isolates resistant to trimethoprim or chloramphenicol. Of the deletions that conferred resistance, they chose to focus on acrB, the very well-studied component of the essential AcrAB-TolC multidrug efflux pump, as well as rfaG, lpxM, and nudB (previously known to confer susceptibility to trimethoprim). They introduced these deletions into an isogenic background, E. coli K-12 MG1655, and documented varying degrees of susceptibility to TMP-SXT, chloramphenicol and others.

Next, they deleted acrB, lpxM, and rfaG from laboratory evolved resistant isolates from a prior study, and measured susceptibilities to a few different antibiotics. They document that acrB deletion confers susceptibility to a variety of antibiotics, consistent with a number of prior studies on acrAB-TolC, and suggesting that the AcrAB-TolC complex likely played a role in conferring resistance in these laboratory evolved isolates, again consistent with prior studies. The authors then challenge these engineered strains with serial passage through high concentrations of trimethoprim to evolve resistance. Here they find (perhaps not surprisingly) that isolates with deleted acrB, lpxM, and rfaG remained susceptible and/or were driven to extinction during the passage experiments at high trimethoprim concentrations that likely substantially inhibited replication (and therefore evolvability).

The authors repeated the experiments at sub-MIC concentrations of antibiotics at which replication would be expected to be permitted, and in these experiments they observed the evolution of resistant (or less susceptible) lineages. They then characterized these lineages by sequencing, and found mutations in some expected targets, including DHFR, the target of trimethoprim, as previously studied by others. They identify a few other targets and interpret their relevance with additional knock out experiments and reasoning based solidly on what is known in the literature. Finally, they performed adaptive evolution experiments with chlorpromazine to inhibit the ArcAB-TolC pump, and found that disruptions in AcrR (the repressor of the acrAB operon) were selected, as expected and consistent with what is known from other studies.

The authors clearly did a lot of careful work for this study, and the manuscript is written in a way in which the experiments and results can be followed throughout. Additionally, the figures are well organized, easy to follow, and information-rich. I had trouble, however, with understanding what exactly the novel or reportable conclusions were, and I found that the qualitative approach to the analysis of the adaptive evolution experiments made quantitative interpretation difficult. Specific comments follow.

Novelty: I think the authors should try to do a more convincing job of explaining exactly what is/are the novel, reportable findings in this work and how the specific novel findings advance our understanding of resistance and evolvability beyond what is known from the >1000 papers published in this area. For instance, the AcrAB-TolC has been extensively studied, and its role in both natural and evolved resistance is well documented and well-understood (as are it's homologs MexAB-OprM, etc). Likewise, many of the other gene targets that were studied are well characterized (for instance acrR, rfaG, lpxM, folA, etc), and the findings the authors report are generally consistent with published literature, or what would be expected, based on what is known. The concept of "resistance proofing" is interesting, but, for instance, the demonstration that isolates with deleted AcrB go to extinction at high concentrations of trimethoprim is not itself a demonstration of a new phenomenon, given the well-understood role that AcrAB-TolC plays in mediating resistance to a variety of chemically-dissimilar antibiotics. If there is something novel that is being presented to support this concept, I think the authors should try to do a better job to explain exactly what the new finding is (beyond the large and very well-organized mutational dataset that is presented).

Quatntiative rigor: The "evolvability" findings at high and low trimethoprim MICs (presumably corresponding to low and high permitted replication rates) are generally consistent with what is well-understood concerning the relationships between replication rates, critical population size, the pool of mutations available for selection, and the rate at which mutations contributing to resistance are discovered by the population. The experiments themselves seem well done, but the analysis has been treated in much more quantitative detail by others over the past decades. If the authors want to make a new argument about selection in these regimes of high and low replication rates, then at the very least the argument will need to be formulated rigorously and quantitatively with actual calculation of generation times at high and low antibiotic concentrations, critical population numbers, and the estimation of mutational pool given the mutation rate per generation. I could not find the starting cell density/exact cell count in the methods. Reference 30 is given, but this critical variable and needs to be included explicitly (the authors will find that the results of their evolvability experiments indeed likely depend critically on the starting cell density, particularly in the experiments with high trimethoprim concentration).

Genome sequencing: Why were the evolved populations expanded in antibiotic free media prior to sequencing? Mutations conferring resistance with high fitness costs may not be represented in the post-expansion population (this has been demonstrated by others), and additionally the proportions of mutations present would not be expected to reflect the original proportions under selection. What are the authors' thoughts on this?

---

## [Decision Letter · Decision Letter 2]

22 Sep 2025

Dear Dr Matange,

Thank you for your patience while we considered your revised manuscript "Antibiotic sensitisation and resistance proofing by impeding pathways of intrinsic resistance in Escherichia coli" for publication as a Research Article at PLOS Biology. This revised version of your manuscript has been evaluated by the PLOS Biology editors, the Academic Editor and the original reviewers.

Based on the reviews, we are likely to accept this manuscript for publication, provided you satisfactorily address the remaining editorial points. Please also make sure to address the following data and other policy-related requests.

a) We routinely suggest changes to titles to ensure maximum accessibility for a broad, non-specialist readership, and to ensure they reflect the contents of the paper. In this case, we would suggest an edit to the title, as follows. Please ensure you change both the manuscript file and the online submission system, as they need to match for final acceptance:

"Impeding pathways of intrinsic resistance in Escherichia coli confers antibiotic sensitisation and resistance proofing"

b) Please add the link to the funding agencies in the Financial Disclosure statement in the manuscript details.

Please supply the numerical values either in the a supplementary file or as a permanent DOI’d deposition for the following figures:

Figure 1BC, 2, 3ABC, 4ABCD, 5, 6ABDEF, 7BC, 8AB

→ It is not clear if the Supplementary File 1 contains this data. If this is the case, please separate them according to the figure for easier access

d) Please cite the location of the data clearly in all relevant main and supplementary Figure legends, e.g. “The data underlying this Figure can be found in S1 Data” or “The data underlying this Figure can be found in https://doi.org/10.5281/zenodo.XXXXX”

e) Please ensure that you are using best practice for statistical reporting and data presentation. These are our guidelines https://journals.plos.org/plosbiology/s/best-practices-in-research-reporting#loc-statistical-reporting and a useful resource on data presentation https://journals.plos.org/plosbiology/article?id=10.1371/journal.pbio.1002128

→ If you are reporting experiments where n ≤ 5, please plot each individual data point.

f) Please make sure that all figures use a colorblind-friendly palette.

We expect to receive your revised manuscript within two weeks.

*Published Peer Review History*

*Press*

Sincerely,

Melissa

Melissa Vazquez Hernandez, Ph.D.

Associate Editor

PLOS Biology

REVIEWERS' COMMENTS

Reviewer #1:

I have read the author replies and revisions and can confirm that the new data and changes to the test satisfactorily addressed my concerns. I can now recommend publication of the paper.

Reviewer #2 (Markus Seeger):

The authors have adequately addressed all my concerns and I consider the paper to be ready to be published in this form.

Reviewer #3:

The authors have extensively revised the manuscript and it is substantially improved. My specific concerns have been sufficiently addressed, and I commend the authors for their excellent work in this interesting study.

---

## [Editor Report · Decision Letter 3]

1 Oct 2025

Dear Nishad,

Thank you for the submission of your revised Research Article "Impeding pathways of intrinsic resistance in Escherichia coli confers antibiotic sensitisation and resistance proofing" for publication in PLOS Biology. On behalf of my colleagues and the Academic Editor, J. Arjan G. M. de Visser, I am pleased to say that we can in principle accept your manuscript for publication, provided you address any remaining formatting and reporting issues. These will be detailed in an email you should receive within 2-3 business days from our colleagues in the journal operations team; no action is required from you until then. Please note that we will not be able to formally accept your manuscript and schedule it for publication until you have completed any requested changes.

PRESS

Sincerely, 

Melissa

Melissa Vazquez Hernandez, Ph.D., Ph.D.

Associate Editor

PLOS Biology
